

# Modeling the Evolution of the Structural Anisotropy of Snow

Silvan Leinss[1], Henning Löwe[2], Martin Proksch[2], and Anna Kontu[3]

[1]Institute of Environmental Engineering, Swiss Federal Institute of Technology in Zurich (ETH), Zürich, Switzerland
[2]Institute for Snow and Avalanche Research SLF, Davos, Switzerland.
[3]Finnish Meteorological Institute FMI, Arctic Research, Sodankylä, Finland.

**Correspondence:** S. Leinss (leinss@ifu.baug.ethz.ch)

**Abstract.** The structural anisotropy of snow that originates from a spatially anisotropic distribution of the ice matrix and the pore space, is a key quantity to understand physical snow properties and to improve their parameterizations. To this end we propose a minimal empirical model to describe the temporal evolution of the structural anisotropy and publish the extensive, calibration dataset consisting of meteorological, radar, and micro computer tomography (CT) data. The dataset was acquired near the town of Sodankylä in Northern Finland. The model is tailored to immediate implementation into common snow pack models driven by meteorological data as its parametrization is solely based on macroscopic, thermodynamic fields. Here we use output data of the physical model SNOWPACK to drive our model. The model implements rate equations for each snow layer and accounts for snow settling and temperature gradient metamorphism, which are taken to be the main drivers of the temporal evolution of the structural anisotropy. The model is calibrated with available time series of anisotropy measurements spanning four different winter seasons. The calibration measurements were obtained from polarimetric radar data which were analyzed with respect to the dielectric anisotropy of snow. From the detailed comparison between simulated anisotropy and radar time series we identify settling as the main mechanism causing horizontal structures in the snow pack. The comparison also confirms temperature gradient metamorphism as the main mechanism for vertical structures. For validation of the model we use full-depth profiles of anisotropy measurements obtained from CT data. The results show that the model can predict the measured CT profiles quite accurately. For depth hoar, differences between modeled anisotropy and the anisotropy derived from exponential correlation lengths are observed and discussed in view of potential limitations.

## 1 Introduction

Deposited snow is a porous material that continuously undergoes microstructural changes in response to the external, thermodynamic forcing imposed by the atmosphere and the underlying soil. Among other microstructural properties, a significant amount of work was recently dedicated to understand the impact of the structural anisotropy which is a key parameter to improve predictions of different snow properties like the thermal conductivity (Izumi and Huzioka, 1975; Calonne et al., 2011; Shertzer and Adams, 2011; Riche and Schneebeli, 2013; Calonne et al., 2014), mechanical (Srivastava et al., 2010, 2016; Wiese and Schneebeli, 2017), diffusive and permeable properties (Zermatten et al., 2011; Calonne et al., 2012, 2014), and also the electromagnetic permittivity (Leinss et al., 2016, and references therein). Especially the thermal conductivity shows a strong dependence on the structural anisotropy (Löwe et al., 2013; Calonne et al., 2014). Depending on snow type, the thermal





conductivity can vary by an order of magnitude at a given density: this variability is discussed with respect to the limits of a completely horizontally and completely vertically structured snow pack (Sturm et al., 1997).

The anisotropy of the snow microstructure is commonly characterized by different variants of geometrical or structural fabric tensors. These can be computed e.g. from mean intercept lengths (Srivastava et al., 2016), contact orientations (Shertzer and Adams, 2011), surface normals (Riche et al., 2013) or other second-order orientation tensors that can be constructed from the two-point correlation function of a two phase medium (Torquato and Lado, 1991; Torquato, 2002). The correlation functions can be evaluated in terms of directional correlation lengths which define characteristic length scales of the microstructure (e.g. Vallese and Kong, 1981; Mätzler, 1997; Löwe et al., 2013). For snow, the microstructure can be obtained by stereology (e.g. Alley, 1987; Mätzler, 2002) or from computer tomography (Schneebeli and Sokratov, 2004).

However the inclusion of the structural anisotropy in current snow pack models is still missing due to i) the lack of a prognostic model for the time evolution of the anisotropy and ii) the lack of in-situ data for validation. Motivated by recent progress of anisotropy measurements using radar (Leinss et al., 2016) as a solution for ii) it is the aim of the present paper to overcome i) and to suggest a minimal, dynamical model tailored to direct use in common, operational snow pack models. The model acts also as a link to connect spatially depth-averaged but temporally high-resolution anisotropy time series from radar with the spatially high-resolution but temporally sparse computer tomography measurements.

The model presented in this paper is based on a simple rate equation which mainly accounts for the influence of snow settling and temperature gradient metamorphism. Each contribution is formulated in terms of macroscopic physical variables like strain rate, temperature and temperature gradient. The magnitude of each contribution is controlled by a free parameter which we calibrated with the radar measurements published in (Leinss et al., 2016). The calibration data consists of radar-measured anisotropy time series covering four winter seasons between October 2009 and May 2013.

The input of our model is based on common state variables provided by detailed snow pack models like SNOWPACK Bartelt and Lehning (2002); Lehning et al. (2002a, b), CROCUS (Brun et al., 1989, 1992) or SNTHERM (Jordan, 1991). Here we used SNOWPACK.

The paper is structured as follows: Section 2 discusses relevant processes which influence the structural anisotropy and casts them into rate equations. Section 3 presents the test site and specifies the field measurement. Section 4 explains the forcing and calibration of the model SNOWPACK and the calibration of the anisotropy model. Section 5 presents the results of the simulated anisotropy profiles and compares them with computer tomographic data. Section 6 discusses capabilities and deficits of the model and indicates possible uncertainties for anisotropy measurements. Section 7 concludes the paper and Section 8 lists used data sets and their availability. The Appendix details the preprocessing of meteorological data and the calibration of SNOWPACK.

Supplementary files provide additional figures about the processing work flow, internal snow temperatures, meteorological data, radiation balance, analysis of SNOWPACK model variants, cost functions for model calibration, density, SSA and correlation lengths derived from CT data, visualizations of snow properties (from SNOWPACK) and additional anisotropy simulations.



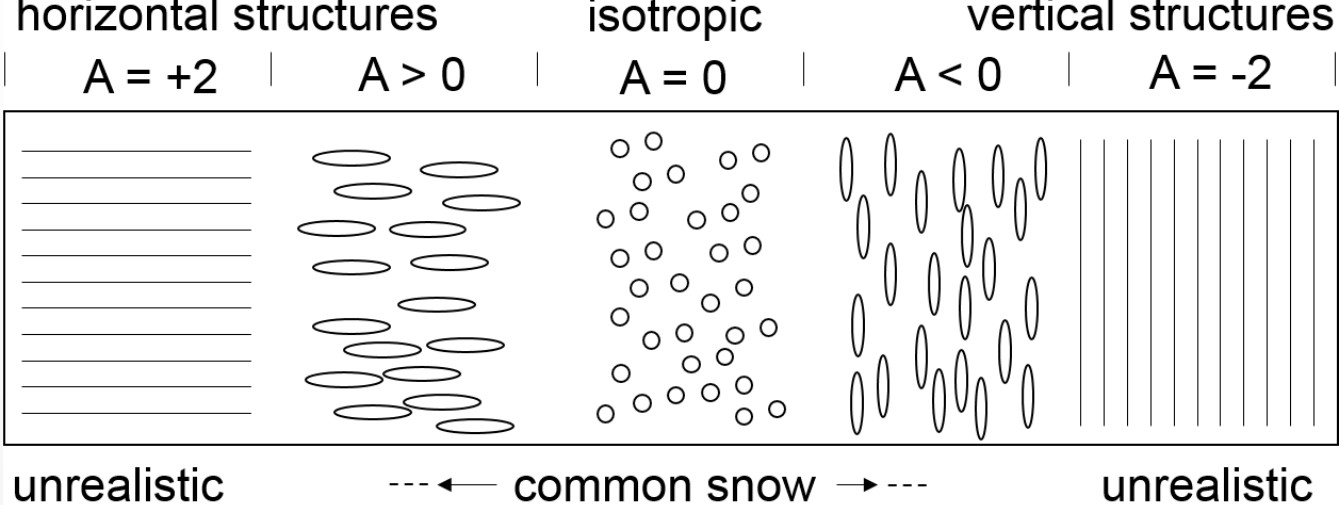

**Figure 1.** Structural anisotropy of different structures according to definition (1). Snow has only a small anisotropy and never reaches the unrealistic cases of horizontal planes or vertical needles.

## 2 A dynamical model for the structural anisotropy

### 2.1 Preliminaries

For quantifying the structural anisotropy, we follow the definition in Leinss et al. (2016) and use the normalized difference of a characteristic horizontal length scale $a_x$ and a vertical length scale $a_z$ and define the anisotropy as

$$A = \frac{a_x - a_z}{\frac{1}{2}(a_x + a_z)}. \tag{1}$$

Different characteristic length scales can be chosen. Commonly, the exponential correlation lengths $a_i = p_{\mathrm{ex},i}$ as defined in (Mätzler, 2002) are used. According to Eq. (1), the structural anisotropy ranges from $A = -2$ (vertical needles) to $A = +2$ (horizontal planes) with $A = 0$ for randomly shaped or spherical particles (visualized in Fig. 1). As detailed in Leinss et al. (2016), the use of a normalized difference is convenient, compared to the anisotropy definition defined via an aspect ratio, $A' = a_z/a_x$, because averaging equally positive and negative values of $A$ result here in isotropy with $A = 0$. The normalized difference defined in Eq. (1) and the frequently used grain size aspect ratio $A'$ are related by

$$A' = \frac{2 - A}{2 + A} \quad \text{or equivalently} \quad A = 2\frac{1 - A'}{1 + A'}. \tag{2}$$

This relation is helpful to compare the anisotropy values of this paper with anisotropy values from other publications which are often given by the aspect ratio $A'$. For weak anisotropies this relation can be approximated as $A' \approx 1 - A$. For snow a common range is $A' \approx 0.75...1.3$ but larger values up to 1.4 might occur (Alley, 1987; Davis and Dozier, 1989; Schneebeli and Sokratov, 2004; Fujita et al., 2009; Calonne et al., 2014). In this range, equally to $A \approx +0.3...-0.3$, the difference $|(1 - A) - A'|$ is less then 5% with respect to $A'$.



For definiteness, we refer to "horizontal structures" when horizontal length scales are larger then the vertical scale, $a_x, a_y > a_z$, hence $A > 0$. Accordingly, "vertical structures" describe snow with larger vertical length scales $a_z > a_x, a_y$ such that $A < 0$.

## 2.2  Evolution of the anisotropy

Quite generally, the anisotropy $A$ in seasonal snow evolves from horizontal structures in fresh snow, over rather isotropic
structures in decomposing rounded grains, to vertical structures under the influence of temperature gradient metamorphism
(Schneebeli and Sokratov, 2004; Calonne et al., 2014) and, at a late stage, returns to isotropy from melt processes. To describe
this evolution we assume the following rate equation

$$\frac{\partial}{\partial t} A(z,t) = \dot{A}_{\text{strain}}(z,t) + \dot{A}_{\text{TGM}}(z,t) + \dot{A}_{\text{melt}}(z,t) \qquad (3)$$

The first term $\dot{A}_{\text{strain}}(z,t)$ accounts for the formation of horizontal structures due to microscopic grain rearrangement in snow
under settling. The second term, $\dot{A}_{\text{TGM}}(z,t)$, accounts for the growth of vertical structures due to temperature gradient meta-
morphism (TGM). The third term, $\dot{A}_{\text{melt}}(z,t)$, causes rounding of grains and a decay of the anisotropy due to melt metamor-
phism. Naturally, in snow all these processes are coupled, so our choice is a pragmatic approximation that seeks an additive
decomposition in terms of these processes.

As common for snow models, we describe the evolution in each layer with a Lagrangian viewpoint where the reference
frame is attached to a material element. Therefore we drop the $z$-dependence in Eq. (3) which would be required for an
Eulerian description where snow layers "sink" through the reference frame fixed in space. Further, we restrict our model to flat
terrain and do not consider any forces acting parallel to the snow layers (in the $x$- or $y$-direction). This implies that gravity and
temperature gradient are strictly applied in the $z$-direction.

## 2.3  Gravitational settling

The first term in Eq. (3), $\dot{A}_{\text{strain}}(t)$, accounts for gravitational settling and densification of snow which apparently creates
horizontal structures which can be observed in polarimetric radar data (Leinss et al., 2014, 2016) as well as in computer
tomographic data (Wiese and Schneebeli, 2017). Densification, in contrast to isotropic contraction by a sintering stress, was
shown to be an anisotropic process: gravity causes an uniaxial squeeze of the snow structure in the z-direction (Fig. 3 and 4 in
Schleef and Löwe, 2013) which increases $A$. The ice matrix is squeezed such that the air pores are filled with above situated
ice grains which move into the gaps by compaction (Theile et al., 2011; Löwe et al., 2011; Schleef and Löwe, 2013), possibly
complemented by rotation of individual fragments of the ice matrix (Löwe et al., 2011), and possibly also by falling of above
situated ice grains into the air-filled gaps (Vetter et al., 2010). In the absence of detailed quantitative work about the anisotropy
of this process we start with the simplest assumption of an affine deformation where all length scales of the structure inherit the
macroscopically imposed scale change from strain. In this case the strain rate and the vertical correlation lengths are related by
$\dot{\epsilon}(t) = \dot{a}_z/a_z$. However, because of the heterogeneous microstructure of snow the assumption of an affine deformation needs





to be mitigated. To account for non-affine effects we introduce an empirical correction factor $\alpha_1$ and hence proceed with

$$\dot{\epsilon}(t) = \frac{1}{\alpha_1} \frac{\dot{a}_z(t)}{a_z(t)}. \tag{4}$$

Then, the anisotropy change rate $\dot{A}(t)$ caused by a strain-induced shortening of the correlation length $a_z$ can be expressed as

$$\dot{A}(t)_{\text{strain}} = \frac{\mathrm{d}}{\mathrm{d}t} A\big(a_z(t), a_x\big) = \left(\frac{\partial A}{\partial a_z}\right)\dot{a}_z(t). \tag{5}$$

Using Eq. (1) and (4) this can be rewritten as

$$\dot{A}_{\text{strain}}(t) = \alpha_1 \dot{\epsilon}(t)\left(\frac{A^2}{4} - 1\right). \tag{6}$$

For large $|A| \to 2$ the term $A^2/4 - 1$ approaches zero and ensures that the anisotropy cannot grow beyond the two extreme values of $A = \pm 2$, even for very large strain rates. However, because the compression of snow is not an affine compression it is unrealistic that large values of $A$ are reached. Therefore, we modify this term and introduce an empirical upper threshold for the anisotropy, $A_{\text{max}} \approx 0.30$, which is based on the maximally observed values for horizontal anisotropies in literature (Leinss et al., 2016; Wiese and Schneebeli, 2017). For negative values of $A$, no modification is applied. This leads to

$$\dot{A}_{\text{strain}}(t) = \alpha_1 \dot{\epsilon}(t)\begin{cases} \left(\frac{A^2}{4} - 1\right) & A \le 0. \\ \left(\frac{A^2}{A_{\text{max}}^2} - 1\right) & A > 0. \end{cases} \tag{7}$$

The strain rate, $\dot{\epsilon} < 0$, ranges between $\dot{\epsilon} \approx -10^{-4}\,\mathrm{s}^{-1}$ for fresh snow with a very low density to $\dot{\epsilon} \approx -10^{-7}\,\mathrm{s}^{-1}$ for old snow of high density (Bartelt and Lehning, 2002). Both, the strain rate $\dot{\epsilon}$ and the $A^2$-terms are always negative, therefore snow settling alway increases the anisotropy $A$.

## 2.4 Temperature gradient metamorphism

The second term in Eq. (3), $\dot{A}_{\text{TGM}}(t)$, accounts for temperature gradient metamorphism (TGM), the most common type of snow metamorphism. Yosida (1955) showed that TGM causes an anisotropic growth of ice crystals which preferably grow into the opposite direction of the heat- and water vapor flux, for both, a horizontal and a vertical heat flux (Yosida, 1955, p. 52–56). The water vapor flux $J_V$ is mediated by diffusion which is driven by a water vapor pressure gradient induced by a temperature gradient. In winter, commonly the soil below the snow pack is warmer than the atmosphere. Therefore, the water vapor pressure is higher at the bottom of the snow pack compared to the cold snow surface. Thus, water molecules diffuse from the bottom up through the ice matrix and form a vertical water vapor flux. Water molecules accumulate on crystals which have a colder temperature than the surrounding air (the bottom side of crystals) and sublimate on the warmer (upper) side of the crystals. This local temperature difference with respect to the surrounding air originates from the higher thermal conductivity of ice which distributes heat faster over the crystal volume. The resulting water transport mechanism has been termed "hand-to-hand" transport by Yosida (1955, p. 31–34). With computer tomography, Pinzer et al. (2012) confirmed this



mechanism and revealed further details: the hand-to-hand transport causes an apparent advection of the ice matrix caused by the downwards motion of (air/ice and ice/air) interfaces that advance by growth or sublimation in the opposite direction of the vapor flux. This leads to a rapid reorganization of the ice matrix with concurrently growing crystals. Pinzer et al. (2012) observed a residence time of water molecules in the ice phase of only few days which makes the idea of slowly growing ice

grains somewhat confusing as *only the "memory" of the grain, encoded in the temporal correlation of the structure, survives* (Pinzer et al., 2012). This continuous reordering of the ice structure under persistent temperature gradients leads to a higher chance for large vertical structures to survive (depth hoar chains) while small structures quickly disappear. To mimic this structural reorganization of the ice matrix, we model the growth of vertical structures proportional to the magnitude of the water vapor mass flux $\dot{A}_{\text{TGM}} \propto |J_{\text{V}}|$.

The absolute value $|J_{\text{v}}|$ is used because vertical structures can grow independent on the sign of $J_{\text{v}}$. In seasonal snow the flux direction is usually positive (upwards) but can be negative in spring, when the (eventually melting) snow surface is warmer than the underlying snow pack, which is likewise the case in perennial snow packs. In contrast, temperature gradients changing their direction on a daily scale seem not to increase the anisotropy but cause a rounding of grains (Pinzer and Schneebeli, 2009). Therefore, we exclude the effect of daily alternating temperature gradients on the anisotropy by averaging temperature

gradients over 24 hours. Larger averaging windows of multiple days did only weakly alter the results. It follows that

$$\dot{A}_{\text{TGM}} \propto |\langle J_{\text{V}} \rangle_{\text{24h}}|. \qquad (8)$$

As indicated in Fig. 1, a perfect needle state has never been observed in a snow pack. Hence we like to restrict the anisotropy to values above a practical minimal anisotropy, $A_{\text{min}}$, which is possible by TGM. By definition, $A_{\text{min}}$ must be larger than -2 (vertical needles). In literature we found that the most negative observed anisotropy values range between $A_{\text{min}} = -0.2$ and

$A_{\text{min}} = -0.35$, corresponding to the range $A' = 1.2...1.4$, Eq. (2), as observed by Fujita et al. (2009): $A' = 1.18$, possibly up to 1.44, Schneebeli and Sokratov (2004): $A' = 1.12$, Alley (1987): $A' = 1.2$, possibly up to 1.4, Calonne et al. (2014): $A' = 1.25$, and $A = -0.3 \pm 0.1$ (CT results, this paper). Calonne et al. (2014) also showed that the anisotropy converged to the value of about $A' = 1.25$ ($A = -0.22$) within three weeks during a constant temperature gradient of $42\,\text{K m}^{-1}$. The observation of a limited growth of anisotropy seems very likely to be related to the limitation of grain size growth as observed by (Sturm and

Benson, 1997) for depth hoar crystals. Despite the differences in the definition of anisotropy metrics used in the examples above, it seems reasonable to empirically limit the growth of vertical structures by a threshold that we set to $A_{\text{min}} = -0.30$.

Additionally, we assume that horizontal structures in fresh snow decay significantly faster than the growth speed of vertical structures in old snow and add an empirical, quadratic weighting function. A faster decay rate of fresh snow compared to old snow partially compensates the fact that any grain size dependence was neglected in the model: the lifetime of small grains in

fresh snow should be significantly shorter than the lifetime of large crystals in old snow.



With the above considerations, we model the second term of Eq. (3) proportional to the vertical water vapor mass flux $J_v$ (kg m$^{-2}$ s$^{-1}$) and the positive prefactor $\alpha_2$.

$$\dot{A}_{\text{TGM}}(t) = -\alpha_2 \frac{|\langle J_v \rangle_{24h}|}{\rho_{\text{ice}} f_\mu(\cdot)} \cdot \begin{cases} \frac{(A - A_{\min})^2}{A_{\min}} & A \geq A_{\min}. \\ 0 & A < A_{\min}. \end{cases} \tag{9}$$

The factor $\alpha_2$ determines the coupling-strength of the right hand side of Eq. (9) and the growth-rate of vertical structures and is later determined empirically based on measured anisotropy time series. On dimensional grounds, we divided the water vapor flux by the density of ice $\rho_{\text{ice}}$ (kg m$^3$) to obtain a velocity. This velocity can be interpreted as the vertical, average ice particle velocity. Divided by a characteristic length scale $f_\mu(\cdot)$ (m) of the microstructure results in the average change rate (s$^{-1}$) of the structural anisotropy. We found, that the model best predicts the measured anisotropy evolution by simply setting $f_\mu(\cdot) = 1$ mm, constant, instead of considering any grain-size dependence. A more physical approach would be to characterize each grain type and size by its potential velocity to transform into vertical structures by a more sophisticated definition of $f_\mu(\cdot)$. Interestingly, any simple, empirical relation could not produce better results compared to the fixed factor $f_\mu(\cdot) = 1$ mm.

The water vapor flux $J_v$ (kg m$^{-2}$ s$^{-1}$) is caused by diffusion of water molecules and originates in concentration gradients in the snow pack. It can be derived by standard thermodynamic relations as done e.g. in (Lehning et al., 2002b). The water vapor mass density, $\rho_v$ (kg m$^{-3}$) is given by the water vapor pressure, $p_S(T)$, and is supposed to be at the saturation point in the pores between the ice crystals. Vapor mass density and vapor pressure are related by the equation for ideal gases and it follows that

$$\rho_v(T) = p_S(T)/(R_V T), \tag{10}$$

where $R_V = R/M_w = 461$ J kg$^{-1}$ K$^{-1}$ is the specific gas constant for water vapor, $M_w = 0.018$ kg mol$^{-1}$ is the molar mass of water and $R = 8.314$ J mol$^{-1}$ K$^{-1}$ is the universal gas constant. The water vapor saturation pressure over ice, $p_S(T)$ can be well approximated using different formulas (Marti and Mauersberger, 1993) and is given in (Bartelt and Lehning, 2002) by

$$p_S(T) \approx p_{0S} \cdot \exp\left[L/R_V \left(T_0^{-1} - T^{-1})\right)\right] \tag{11}$$

with the latent heat of ice sublimation $L = 2.8$ MJ kg$^{-1}$, the Triple point pressure $p_{0S} = 611.73$ Pa, and Triple point temperature $T_0 = 273.16$ K of water.

The vertical water vapor mass flux $J_v$ is determined by Fick's law applied to the vapor mass density $\rho_v(T)$ and seems to be almost independent on grain size or microstructure (Pinzer et al., 2012, Fig. 11). Because the saturation pressure depends only on temperature, the vertical water vapor mass flux can be written in terms of temperature $T$ and temperature gradient $\frac{\partial T}{\partial z}$ according to (Lehning et al., 2002b):

$$J_v\left(T, \frac{\partial T}{\partial z}\right) = -D_{\text{vs}} \frac{\partial \rho_v}{\partial z} = -D_{\text{vs}} \frac{\partial \rho_v(T)}{\partial T} \frac{\partial T}{\partial z} \tag{12}$$

$$= -D_{\text{vs}} \cdot \rho_v(T) \cdot \left[\frac{L}{R_v T^2} - \frac{1}{T}\right] \frac{\partial T}{\partial z} \tag{13}$$

The effective diffusion constant for water vapor in snow, $D_{\text{vs}}$, is close to the diffusion constant in air, $D_{\text{v,air}} = 2.1 \cdot 10^{-5}$ m$^2$ s$^{-1}$, (Massman, 1998) and ranges between 1 and $10 \cdot 10^{-5}$ m$^2$ s$^{-1}$ (Sokratov and Maeno, 2000; Colbeck, 1993), see also review in (Pinzer et al., 2012). We assumed a constant diffusion constant, $D_{\text{vs}} = 2 \cdot 10^{-5}$ m$^2$ s$^{-1}$.





Extremely large temperature gradients could naturally occur at the snow surface under extreme conditions but we do not expect that the anisotropy will grow proportionally at such extreme rates. Additionally, extreme temperature gradients could wrongly occur in simulated data. To exclude such temperature gradients, we set a maximum threshold for simulated temperature gradients of $|\Delta T/\delta z| \leq 200\,\mathrm{K\,m^{-1}}$.

## 5  2.5  Melt metamorphism

Despite a lack of calibration data for anisotropy change under melt metamorphism we implement here a simple model. Calibration data is almost not existent, because the 9–18 GHz radar, which we used to measure the anisotropy evolution and to calibrate our model, cannot penetrate wet snow. Only one event in April 2012 is available where the snow refroze after strong surface melt occurred. Additionally, observational data and models to predict wet snow metamorphism are still rudimentary

and except for the model and references in (Lehning et al., 2002a) and (Brun et al., 1992) we could not find any detailed studies. Similar to their given rate equations we model the anisotropy decay due to melt metamorphism as

$$\dot{A}_{\mathrm{melt}} = -\alpha_3 A \cdot \theta_{\mathrm{w}}^{\mathrm{v}\,3} \tag{14}$$

with the empirical constant $\alpha_3 = 2 \cdot 10^{-3}\,\mathrm{day^{-1}}$ and the liquid water volume fraction $\theta_{\mathrm{w}}^{\mathrm{v}}$ in vol.%. The term $\dot{A}_{\mathrm{melt}}$ has always the opposite sign of $A$ and causes therefore always a rounding of grains and a decay of anisotropic structures.

## 15  3  Datasets and testsite

All data for model forcing, calibration and validation were acquired at or close to the test site "intensive observation area" (IOA), located 5 km south of the town of Sodankylä in northern Finland. The IOA is shown in Fig. 2. Table 1 lists all used measurements, sensors and locations. The measurements were supported by the Nordic Snow Radar Experiment NoSREx-I to -III (Lemmetyinen et al., 2013, 2016).

The anisotropy model was forced by depth-resolved snow properties simulated by the model SNOWPACK (v. 3.4.5). SNOWPACK was driven by meteorological data mainly from the IOA. Precipitation and wind velocity were measured 600 m north of the IOA at the automatic weather station (AWS). The radiation balance was measured close to the AWS at the sounding station (short wave) and at the radiation tower (long wave).

SNOWPACK was calibrated by snow height (IOA, AWS, meteorological mast) and by snow temperature. Snow temperature

was measured at the meteorological mast, 180 m east of the IOA, with an array of horizontal temperature sensors spaced 10 cm vertically in the snow pack (Fig. 3).

Free parameters of the model ($\alpha_1, \alpha_2, \alpha_3, A_{\mathrm{ini}}$) were estimated by radar-measured time series of depth-averaged anisotropy measurements. The measurements are described in detail in (Leinss et al., 2016) and are based on polarimetric phase measurements from which we derived the dielectric birefringence of the snow pack at microwave frequencies. The measurements were

done with the SnowScat radar instrument which was developed and built to analyze the backscatter intensity of snow between 9.2 and 17.8 GHz (Werner et al., 2010; Lemmetyinen et al., 2016), ESA ESTEC contract 42000 20716/07/NL/EL (available on







**Figure 2.** Picture of the intensive observation area (IOA) where field-, radar-, and most meteorological data were acquired. Abbreviations are explained in Table 1. Anisotropy validation profiles were extracted at the locations CT-1, CT-2a/b, CT-3, and CT-4. The tower-based SnowScat instrument measured the depth-averaged anisotropy every four hours over the area "sector 1" (Leinss et al., 2016). It also measured snow water equivalent (SWE) in combination with the gamma water instrument, GWI (Leinss et al., 2015). Additional meteorological data were measured at the meteorological mast 180 m east of the IOA and at the automatic weather station (AWS) 600 m north of the IOA.





request from ESA). Tilting and rotation of the radar antennas allowed for anisotropy measurements at both, different incidence- and different azimuth-angles of sector 1 of the IOA.

For validation, we compared the modeled anisotropy profiles with computer tomographic measurements of the snow microstructure. Vertical anisotropy profiles were computed from the microstructure which was sampled during four field visits to
the location CT-1, -2, -3, and -4 shown in Fig. 2.

The following subsections provide intermediate details of the retrieval, preprocessing, and filtering of the ground measurements. More details are provided in Appendix A1 and A2. Plots of SNOWPACK input, output and control data are provided in the supplementary material.

## 3.1   Meteorological input data

For definition of the snow-atmosphere boundary conditions, SNOWPACK requires the following meteorological input data: air temperature (TA), soil temperature (TSG), relative humidity (RH), wind speed (VW), wind direction (DW), incoming short wave radiation (ISWR) and/or reflected (outgoing) short wave radiation (OSWR), incoming long wave radiation (ILWR) and/or snow surface temperature (TSS), precipitation (PSUM) and/or snow height (HS) and optionally the precipitation phase (PSUM_PH). For monitoring purposes, up to five internal snow temperature measurements (TS1, ..., TS5) at different heights
can be provided for comparison with modeled snow temperatures. Most of these quantities were measured by more than one sensors at the IOA and nearby sites (Table 1). To provide physically correct and consistent conditions, the meteorological data were filtered, combined, and interpolated if gaps could not be filled with equivalent datasets. Preprocessing details of meteorological data are provided in Appendix A1. Plots of both measured raw data and filtered and pre-processed data (SNOWPACK input), are provided in the supplementary figures S2–S9. SNOWPACK additionally filters and pre-processes the input data and
provides them for control (supplementary figures S10–S13).

## 3.2   Definition of underlying soil

For the lower boundary condition, SNOWPACK requires a description of at least one soil layer. To define precisely the temperature of the soil-snow interface we defined a single, 5 cm thin soil layer which lower temperature (TSG) was determined by the average of four soil temperature sensors at -5 cm and -10 cm (sensor arcsoil at meteorological mast) and two measure-
ments at -2 cm depth (sensor SMT at IOA). For soil moisture we averaged data from six sensors, two from the meteorological mast (arcsoil: -5 cm, -10 cm) and four from the IOA (SMT: two locations each at -2 cm and -10 cm). For the definition of soil properties at the start of the simulation we provided soil temperature and moisture as the average over one week around the simulation start time (1st of September).

The soil composition is described in (Lemmetyinen et al., 2013) as very fine mineral soil composed of 70% sand, 1% clay
and 29% silt. For this mineral soil, we assumed a solid volume fraction of 75% and zero ice fraction in autumn. We estimated a density 1800 kg m$^{-3}$, a heat conductivity of 1.5 W m$^{-1}$ K$^{-1}$ (from ToolBox (2003a)), and a heat capacity of 1000 J kg$^{-1}$ K$^{-1}$ (from ToolBox (2003b)). A soil albedo of 0.2 was determined from the ratio of incoming and reflected short wave radiation data (sensor: CM11 at sounding station).





**Table 1.** List of data sources for model input, calibration and validation. Sites are given with coordinates, below follow sensor abbreviations and full sensor names, or data set abbreviation and type of measurements.

| | |
|---|---|
| **Intensive observation area (IOA):** | 67.36185°N, 26.63355°E |
| SDAT1 | Sensor for snow height and air temperature |
| SMT A,B | Two sensors for soil moisture (at -2, -10 cm), |
| | and soil temperature (at -2 cm) |
| SnowScat | SnowScat instrument (SSI), tower-based radar for |
| | depth-averaged anisotropy measurements |
| | and snow water equivalent, SWE. |
| CT-no. | Snow profile no.1...4, analyzed by computer |
| | tomography, CT: (anisotropy validation data) |
| CT-1 | Profile 1, sampling date: 03 March 2011 |
| CT-2a/b | Profile 2a/b, sampling date: 21 December 2011 |
| CT-3 | Profile 3, sampling date: 01 March 2012 |
| CT-4 | Profile 4, sampling date: 28 February 2013 |
| GWI | Gamma Water Instrument |
| | (SWE measurement by gamma ray absorption) |
| Distr | Distrometer: precipitation classification, |
| | precipitation phase: liquid, solid |
| Snow pit | Snow pit for snow density, SWE, grain size |
| | (manual measurements) |
| | |
| **Meteorological mast (arcmast):** | 67.36205°N, 26.63723°E |
| arcsnow | Snow height, air temperature (1 m above ground), |
| | snow temperature at 10, 20, ..., 110 cm height |
| arcsoil | Soil moisture, soil temperature at -5, -10,...-50 cm |
| SDvar | Snow height variability course (7 x snow height) |
| | |
| **Automatic weather station (AWS):** | 67.36662°N, 26.62898°E |
| | Snow height, air temperature (2 m above ground), |
| | wind speed, wind direction, precipitation, |
| | relative humidity |
| | |
| **Sounding station (near AWS):** | 67.36660°N, 26.62975°E |
| CM11 | Kipp&Zonen sensor CM11, 305–2800 nm, |
| | incoming short wave (global), ISWR, |
| | outgoing short wave, OSWR |
| | |
| **Radiation tower (near AWS):** | 67.36664°N, 26.62825°E |
| CG4 | Kipp & Zonen sensor CG4, 4500–42000 nm, |
| | incoming long wave radiation, ILWR, |
| | outgoing long wave radiation, OLWR |

## 3.3 Snow temperature

The internal snow temperature was measured by an array of 11 horizontally oriented temperature sensors located at 10, 20, ..., 110 cm above the ground (Fig. 3). Unfortunately, for this configuration with all sensors attached to the same support stick, we cannot exclude that some air-filled gaps occurred between the sensor elements. Furthermore, it was reported for another,





**Figure 3.** Snow temperature was measured at every 10 cm height by an array of horizontally oriented temperature sensors at the meteorological mast.

similar sensor configuration that the sensor configuration interfered with snow accumulation and caused the formation of a pit (up to 30 cm deep) in the snow around the sensor. For the sensor used here, such measurement errors can be detected by comparing the lowest snow temperature (at +10 cm above ground) with the soil temperature (see Figs. A5 and A6). For a deep, well insulating snow pack, both temperatures should not vary more than a few K. Manual snow temperature measurements provide an additional validation source for the sensor array measurements.

### 3.4 Anisotropy determined by polarimetric radar

For calibration of the anisotropy model we used depth-averaged anisotropy time series which were obtained from polarimetric radar measurements acquired by the ground based radar instrument SnowScat. The method is described in detail in (Leinss



et al., 2016). Below, we briefly summarize the method. For technical details of the instrument see (Werner et al., 2010) and also (Leinss et al., 2015) where the instrument was used to determine the snow water equivalent (SWE) at test site IOA.

Microwaves with sufficiently long wavelength penetrate the snow pack with negligible scattering losses and accumulate a signal delay due to the refractive index of snow. When the snow pack has a spatially anisotropic structure (i.e. preferably

horizontal or vertical structures) and the electric field has horizontal and vertical components, the signal delay depends on polarization. The average anisotropy of the snow pack determined by radar, $A_{\mathrm{avg}}^{\mathrm{CPD}}$, can therefore be derived from the signal delay difference of two perpendicular to each other polarized radar echoes. This signal delay difference can precisely be measured interferometrically by analyzing the phase difference between the two orthogonally polarized microwaves. This phase difference is called the co-polar phase difference, CPD (Leinss et al., 2016). When this interferometric method is applied

at sufficiently high frequencies (10–20 GHz) it is possible to determine $A_{\mathrm{avg}}^{\mathrm{CPD}}$ with an accuracy of a few percent. The frequency limits are determined such that the radar penetration depth in snow is sufficiently height (upper limit), that the measurable phase accuracy is much smaller than the total CPD, and that penetration into soil (and polarimetric effects of soil) are negligible (lower limit). About 3200 anisotropy measurements with a temporal resolution of 4 hours were acquired during the four winter seasons 2009–2013. Because microwaves frequencies above 10 GHz have almost no penetration into wet snow, the anisotropy

during snow melt could not be measured.

### 3.5  Anisotropy determined by computer tomography

For validation of different active and passive microwaves experiments conducted during the NoSREx campaigns, snow samples comprising almost complete vertical snow profiles were extracted in the field and conserved for a later analysis by means of micro computer tomography (CT). The profiles contain some gaps of a few cm where the samples were not overlapping or

sample taking was not possible due to very soft fresh snow, ice crusts or large fragile depth hoar crystals. The profiles were sampled at the five locations, CT-1, CT-2a, CT-2b, CT-3, and CT-4, shown in Fig. 2. Sampling dates are listed in Table 1. The snow samples were analyzed at the WSL-Institute for Snow and Avalanche Research SLF in Switzerland.

We analyzed the CT data with respect to the structural anisotropy of snow. Other derived snow parameters have already been published in (Proksch et al., 2015). Here we briefly summarize the methodology of sample preparation and processing of the

CT-data: for transportation from Finland to the cold lab at SLF, the snow samples were cast using Diethyl-Phthalate (DEP) as described in (Heggli et al., 2009). In the cold lab, the samples were scanned with a nominal resolution (voxel size) ranging from 10 μm for new snow to 20 μm for depth hoar. The resulting 3D-gray-scale images were filtered using a Gaussian filter (sigma = 1 voxel length, total filter kernel width = 5 voxel lengths). The smoothed images were then segmented into binary snow/air images. For segmentation, an intensity threshold was chosen at the minimum between the DEP peak and the ice peak

in the histograms of the gray-scale images. From the binary images two-point correlation functions were calculated according to Löwe et al. (2013) for each direction and the corresponding correlation lengths, $p_{\mathrm{ex},x}$, $p_{\mathrm{ex},y}$, and $p_{\mathrm{ex},z}$, were derived as described by Mätzler (2002). Examples of the analyzed snow samples are shown in (Leinss et al., 2016, Fig. 14 and 15).





The anisotropy determined by computer tomography, $A^{\mathrm{CT}}$, is defined analogue to Eq. (1). Because of the symmetry in the $x$-$y$-plane, the correlation lengths $p_{\mathrm{ex},x}$ and $p_{\mathrm{ex},y}$ were averaged:

$$A^{\mathrm{CT}} = \frac{0.5(p_{\mathrm{ex},x} + p_{\mathrm{ex},y}) - p_{\mathrm{ex},z}}{\frac{1}{2}\left[0.5(p_{\mathrm{ex},x} + p_{\mathrm{ex},y}) + p_{\mathrm{ex},z}\right]}. \tag{15}$$

In contrast to the depth-averaged radar measurements used for model calibration, the anisotropy, $A^{\mathrm{CT}}$, has a vertical resolution
of 1–2 mm and allows for validation of the simulated anisotropy profiles.

### 3.6 Snow classification and NIR images

Datasets of manual snow type classification have been acquired on a weekly basis. In addition for several dates near-infrared (NIR) images of the snow structure are available. For each snow image the ratio relative to a reference image of a Styrofoam panel was calculated. The ratio-images were used to cross-check snow type classification as well as the CT data. They were
also considered for interpretation of the simulated results. The images are shown in Fig. 8.

## 4  Methods: model forcing and calibration

The proposed anisotropy model is designed for immediate implementation into snow pack models which provide the following variables for each layer of snow: snow temperature $T$, vertical snow temperature gradient $\partial T/\partial z$, and strain rate $\dot{\varepsilon}$. The software SNOWPACK (Bartelt and Lehning, 2002; Lehning et al., 2002a, b) provides these parameters but does not consider
the structural anisotropy of snow. To keep the implementation simple enough, we post-processed the output of SNOWPACK and did not intent to feed the anisotropy back into SNOWPACK.

### 4.1  SNOWPACK: calibration and configuration

SNOWPACK provides a variety of settings to adjust for the local environment and to configure the simulation. Additionally, the radiation balances required some calibration because it was not directly measured over the IOA. To best replicate measured
snow height and temperatures we run for all four seasons more than 5000 simulations with different settings and graded the accuracy of the simulation results by comparison of simulated snow height and snow temperature with measured snow height and temperature (TS1, ..., TS5). To avoid systematic deviations of SWE or snow density we first run SNOWPACK driven by calibrated precipitation (details in Appendix A1). Then, we run the best 300 simulations again but with enforced snow height; for a sanity check we verified the simulated SWE. Table 2 summarizes the most important settings which significantly
improved the simulation results. Details about the quantitative comparison of snow height and snow temperature and the definition (grading) of the "best" simulations are described in Appendix A3.

Simulation results significantly improved by setting SNOW_EROSION = TRUE and WIND_SCALING_FACTOR ≈ 2. Results also improved by setting ATMOSPHERIC_STABILITY to NEUTRAL. Little difference was found between a fixed threshold for the precipitation phase (THRESH_RAIN, Table 2) and estimation of the precipitation phase from distrometer
data (Appendix A1).




**Table 2.** Most relevant settings for SNOWPACK which provided the best results.

| | |
|---|---|
| SNOW_EROSION | = TRUE |
| WIND_SCALING_FACTOR | = 2.0...2.5 |
| ATMOSPHERIC_STABILITY | = NEUTRAL |
| THRESH_RAIN | = 0.7...1.2°C, (or PSUM_PH) |
| ISWR | = ISWR × 0.65...0.95 |
| ILWR | = ILWR × 0.93...0.97 |
| SW_MODE | = INCOMING, (BOTH) |

Tree canopy was not considered (CANOPY = FALSE) because the test site was not covered by trees. Still, surrounding trees could have affected the radiation balance. Radiation data was calibrated by multiplication with constant factors and selection of the best simulation results. Incoming short wave radiation (ISWR) was multiplied by 0.65...0.95 (Table 2) which agrees with the fact that the IOA was partially shadowed by trees but short wave radiation was measured on a tower above the trees. Outgoing

short wave radiation (OSWR) was not used but estimated by SNOWPACK based on the simulated albedo (SW_MODE = INCOMING instead of BOTH). As expected, the calibration factor for incoming long wave radiation (ILWR) is close to one because ILWR mainly results from diffuse reflected radiation by cloud cover. Outgoing long wave radiation was not used by SNOWPACK.

## 4.2   Calibration of the anisotropy model

Several free parameters of the model had to be determined. The values of $A_\mathrm{min}$ and $A_\mathrm{max}$ have already been defined based on literature values in Sect. 2.3 and 2.4. The initial anisotropy of new snow, $A_\mathrm{ini}$, is discussed in Sect. 4.4. The remaining three coupling parameters, which describe the effect of settling ($\alpha_1$), TGM ($\alpha_2$), and melt metamorphism ($\alpha_3$), were determined as follows: $\alpha_1$ and $\alpha_2$ were estimated iteratively by minimization of the error (cost function, Eq. A1) between the modeled time series for the anisotropy $A_\mathrm{avg}^\mathrm{mod}$ (depth-averaged) and the anisotropy $A_\mathrm{avg}^\mathrm{CPD}$, measured by radar. For calibration we only used radar

measurements which were considered reliable enough, i.e. when the snow was deep enough such that the radar measurements showed a standard deviations $\sigma(A_\mathrm{avg}^\mathrm{CPD}) < 0.05$ and when no wet snow perturbed the transmissive radar measurements. Details about optimization and the cost function are provided in Appendix A4.

Finally, upon determination of $\alpha_1$ and $\alpha_2$, the parameter $\alpha_3$ was estimated manually, mainly, by considering the melt-event on 11. April 2012. This melt event occurred simultaneously with snow fall which, in spite of settling, did only marginally

increase the radar-measured anisotropy. The melt metamorphism term counteracts here the settling-induces anisotropy increase by modeling rounding of anisotropic ice grains. The parameter $\alpha_3$ is only weakly constrained by measurements a few days around this event. Unfortunately, during the entire melt period, anisotropy measurements with radar are not possible (Sect. 2.5).





### 4.3 Model calibration for different snow packs

To specify the uncertainty for the free parameters $\alpha_1$ and $\alpha_2$ we did not only globally evaluate the cost function over all four seasons, but also determined $\alpha_1$ and $\alpha_2$ independently for each season. The results of this analysis show whether the parameters $\alpha_1$ and $\alpha_2$ must be adjusted for every different snow pack or if they could be universally valid, at least for the four seasons of
Finish snow.

Furthermore, the cost function was evaluated for the ensemble of the best 300 SNOWPACK simulations with slightly different configuration settings (scaling of radiation balance, rain threshold, wind scaling factor, short wave reflected radiation based on albedo simulation or measurements, precipitation phase estimation). However, all 300 simulations had the following settings in common: snow height was enforced, neutral atmosphere, snow erosion was allowed. The quality of this ensemble
of simulations, assessed by comparison to measured snow height and snow temperature, is shown in histograms in the supplementary Fig. S16. This sensitivity analysis provides an estimate how sensitive the parameters $\alpha_1$ and $\alpha_2$ are with respect to slightly changing snow conditions in different ensemble members.

### 4.4 Initial conditions

Linking the output of SNOWPACK to our anisotropy model required specification of the initial anisotropy for fresh snow and
proper handling for merging of snow layers.

The anisotropy of fresh snow was set to $A_{\mathrm{ini}} = 0.05$. $A_{\mathrm{ini}}$ is expected to be positive but close to zero, as each fresh snow layer has already settled during accumulation. The initial value of 0.05 seems to be realistic in view of a strain rate of $10^{-5}\,\mathrm{s}^{-1}$ of fresh snow during the input data sampling interval of one hour. Furthermore, it is likely that snow flakes align preferably horizontally by gravity at the time of deposition. This assumption is supported by observations where dendrites were only
found with horizontal orientation in artificial snow (Löwe et al., 2011) as well as in natural snow (Mätzler, 1987, Fig. 2.15). For the initial anisotropy, we neglected any temperature dependence due to lack of representative data. Stronger cohesion between crystals at temperatures close to zero could lead to a more isotropic structure (but with faster settling) compared to cold temperatures were crystals align according to gravity without being influenced by stronger cohesion forces or settling. A temperature dependence for the shape of snow crystals growing in the atmosphere could also influence the initial anisotropy.

SNOWPACK merges two adjacent snow layers when they have similar properties and when their thickness falls below a certain threshold. To keep track of the anisotropy evolution of merged layers, we wrote an algorithm to detect when snow layers get merged. The anisotropy of a merged layer is defined by the average anisotropy of the two original layers weighted by their thickness.

### 4.5 Implementation

The rate equations were solved with a simple explicit Euler method. The classical Runge-Kutta algorithm was also implemented but did not show visible difference to the explicit Euler method which was used for computational efficiency: A single iteration, simulating about 600 000 snow elements takes about 1.0 s (1.9 s for Runge-Kutta) summing up to about 45 s to determine the





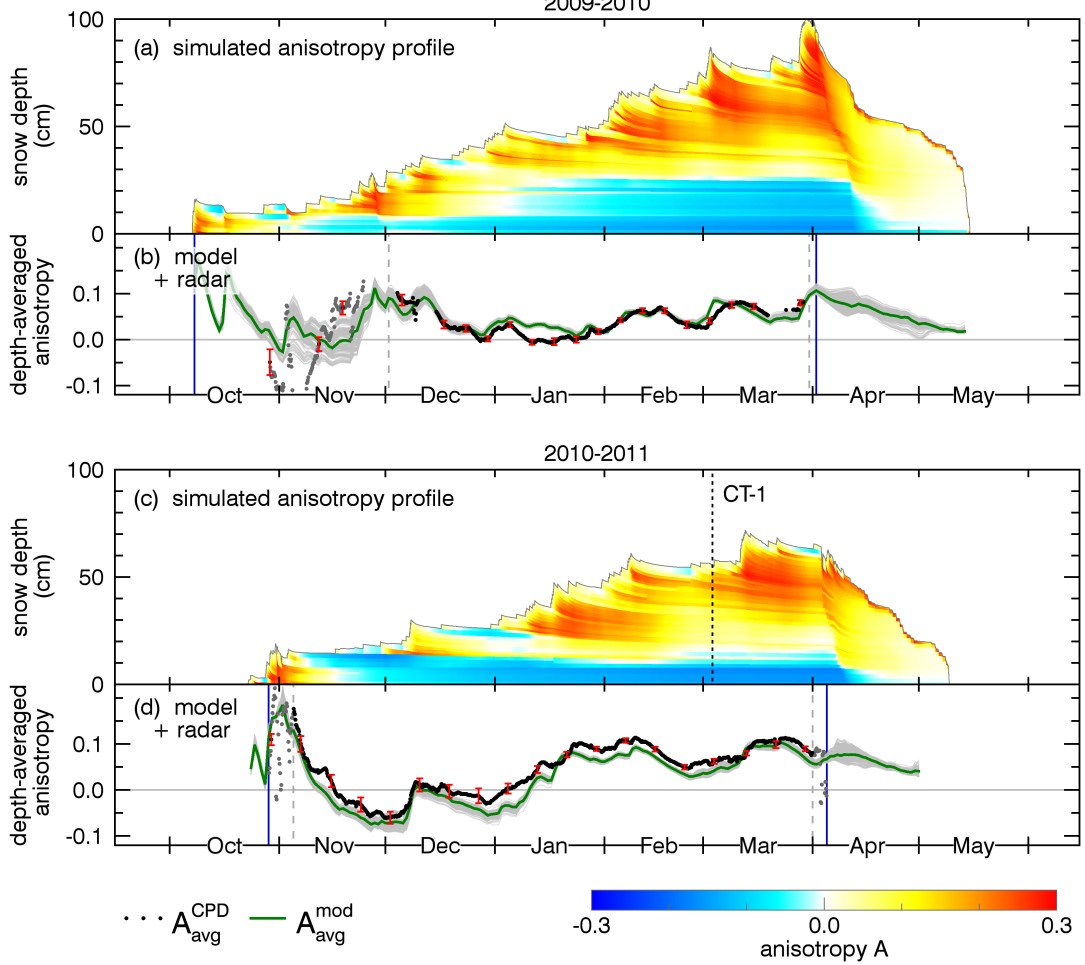

**Figure 4.** Simulation results for the structural anisotropy of the first two seasons 2009/2010 and 2010/2011. (a), (c), in color: time- and depth-resolved modeled anisotropy based on post-processed SNOWPACK data. The dashed line, labeled with CT-1, indicates the sampling date of the CT validation data. Time-series panels (b), (d): depth-averaged anisotropy of the model $A_{avg}^{mod}$ (green: ensemble median, gray: ensemble members) compared with radar-measured anisotropy $A_{avg}^{CPD}$. Dashed gray lines bound the radar measurements (black dots) used for model calibration. Gray dots indicate radar measurements excluded from calibration because of a large standard deviation (red error bars).

parameters $\alpha_1$ and $\alpha_2$ for a single SNOWPACK ensemble member representing four winter seasons. The entire code was implemented in IDL on a PC with Intel Xeon CPU E3-1270 V2 @ 3.50GHz, 4 cores.

# 5   Results

The simulated evolution of the anisotropy is shown in Fig. 4 and Fig. 5. In the upper panels, (a) and (c), the anisotropy of each
5   snow layer is shown in color: yellow and red colors indicate horizontal structures and blue colors indicate vertical structures.



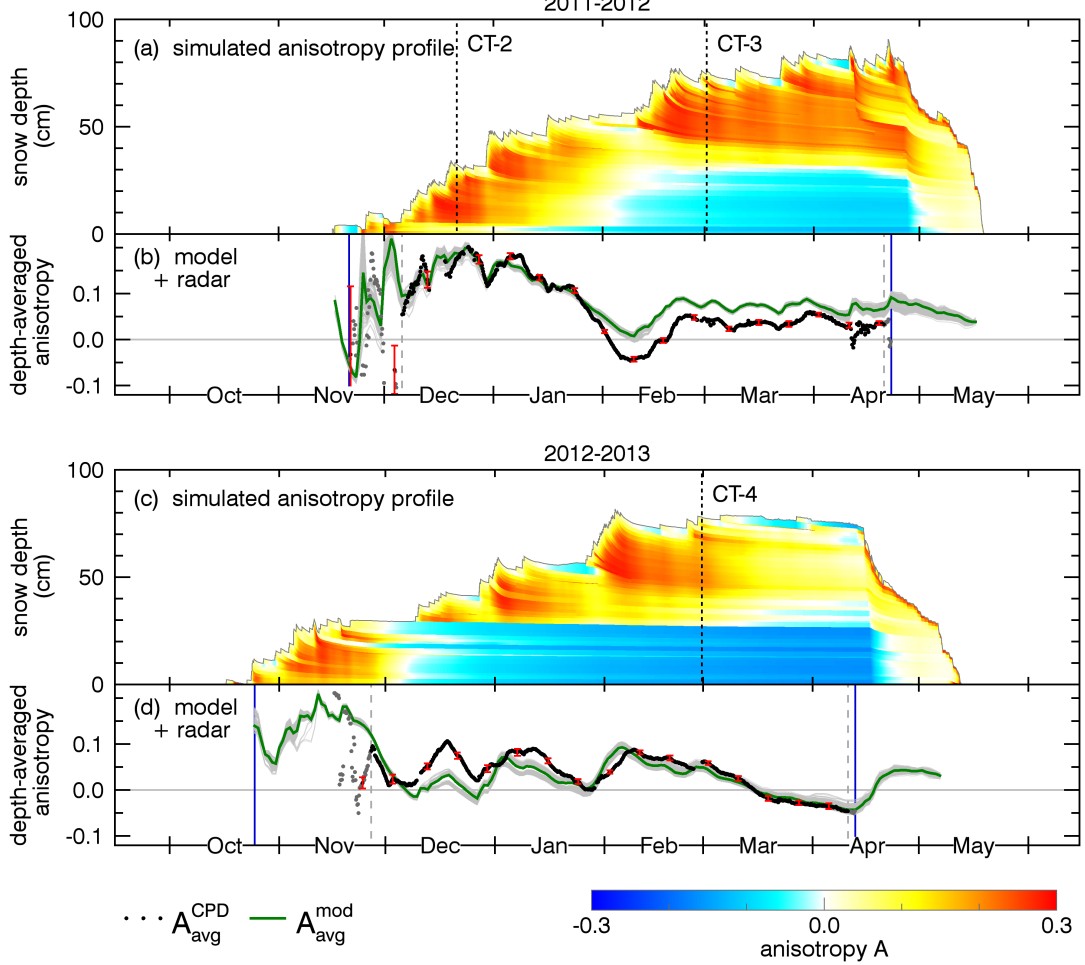

**Figure 5.** Simulation results for the seasons 2011/2012 and 2012/2013. Labels CT-2, CT-3, and CT-4, indicate the sampling dates of the CT validation data. For further details see caption of Fig. 4.

The shown profiles are based on input data from the "best" SNOWPACK run, compared to all other ensemble members (Appendix A3). The simulated anisotropy values range approximately between A = -0.2 and A = +0.3. The dates when the CT profiles were obtained in the field are indicated by vertical black dashed lines labeled with CT-1, -2, -3, and -4.

Below the color-coded depth-resolved profiles, the panels (b) and (d) show time series of the depth-averaged anisotropy. The radar-measured anisotropy, $A_{avg}^{CPD}$, is colored in black, measurements not used for model calibration are shown as gray dots. Red error bars indicate the standard deviation of radar measurements with different incidence and azimuth angles. The depth-average of the modeled anisotropy, $A_{avg}^{mod}$, is shown in green and gray lines: the green line is the median of the 300 best SNOWPACK runs which are shown as the ensemble of gray lines in the background. Begin and end of the dry snow period





are indicated by vertical blue lines. Dashed gray lines bound the period of radar measurements considered reliable enough to determine the free parameters $\alpha_1$ and $\alpha_2$.

It is remarkable that modeled and radar-measured anisotropy, $A_{\mathrm{avg}}^{\mathrm{mod}}$ and $A_{\mathrm{avg}}^{\mathrm{CPD}}$, show a highly consistent trend: the model is able to catch even small details of the radar measured anisotropy time series. Nevertheless, over some periods, especially in the

season 2011/2012, Fig. 5(b), longer systematic deviations persist. For completeness, correlation measures between modeled and measured anisotropy time series are listed in Table 3.

**Table 3.** Correlation between simulated and radar-measured anisotropy, $A_{\mathrm{avg}}^{\mathrm{CPD}}$ and $A_{\mathrm{avg}}^{\mathrm{mod}}$, for each season and all seasons together (last row). To quantify the model quality the following measures are listed: $r$ = Pearson's correlation coefficient, NS = Nash-Sutcliffe model efficiency coefficient, RMS = root mean square difference. The measures are given for the best SNOWPACK simulation (left) and for the ensemble median of the best 300 simulations (right).

|  | best SNOWPACK sim. | | | ensemble median | | |
|---|---|---|---|---|---|---|
| season | $r$ | NS | RMS | $r$ | NS | RMS |
| 2009/2010 | 0.88 | 0.69 | 0.016 | 0.85 | 0.66 | 0.016 |
| 2010/2011 | 0.97 | 0.85 | 0.021 | 0.98 | 0.81 | 0.023 |
| 2011/2012 | 0.96 | 0.67 | 0.036 | 0.96 | 0.70 | 0.035 |
| 2012/2013 | 0.75 | 0.30 | 0.033 | 0.81 | 0.56 | 0.026 |
| 2009–2013 | 0.84 | 0.69 | 0.027 | 0.86 | 0.72 | 0.026 |

From the simulated anisotropy profiles it is evident that snow layers at the bottom of the snow pack always shows a vertical structures (blue, $A < 0$) while the upper snow layers which are stronger affected by snow settling show generally horizontal structures (yellow and red, $A > 0$). However, the snow surface shows a more isotropic (and sometimes an even vertical) struc-

ture compared to the underlying upper snow layers which experienced more overburden pressure. The occasionally appearing vertical structures at the snow surface are expected because of strong temperature gradients at the surface, especially during clear-sky winter nights. During such conditions, TGM transforms the top layers faster than intermediate layers.

A small but very interesting detail of both, the model and the radar measurements is that the anisotropy does not grow instantaneously with accumulating fresh snow but shows a short delay of a few days. This is clearly visible in both, the

anisotropy profiles and the depth-averaged data, after intense snow fall events, e.g. in March 2010, March 2011, and February 2013. The delay results from the fact that it is the settling of fresh snow which dominantly increases the anisotropy while fresh snow itself only weakly increases the anisotropy. The delay in radar measurements seems to be even more pronounced that the simulated results. Such a delay of about 2–4 days in average was also observed in (Leinss et al., 2016, Sect. 5.4). Compared to the fast evolution of fresh snow, the model predicts a much slower evolution for old snow with pronounced vertical structures.

## 5.1   Validation with computer tomography

The vertically resolved anisotropy profiles from computer tomography make it possible to use the CT data for validation. Figure 6 shows simulated anisotropy profiles as blue lines and the CT-based anisotropy as gray dots with a black line indicating the



**Figure 6.** Vertical profiles of the field-measured anisotropy, $A^{\mathrm{CT},p_{\mathrm{ex}}}$, determined by computer tomography from exponential correlation lengths (gray dots; black line: 5 cm running mean). Green line: anisotropy calculated from the correlation length $p_c$ (5 cm running mean). Blue line: simulated anisotropy, $A^{\mathrm{mod}}$. Horizontal black dashed line: field-measured snow height (HS). The right axis shows layer classification according to (Fierz et al., 2009). The four ticks at the top are the mean anisotropy values of CT($p_{\mathrm{ex}}$), CT($p_c$), radar (CPD), and modeled anisotropy (mod).



5 cm running mean of the CT-based anisotropy. The CT-based anisotropy is derived from exponential correlation lengths $p_{ex}$ according to Eq. 15.

Additional to the anisotropy based on $p_{ex}$, the anisotropy could also be calculated from other correlation lengths. Being aware, that $p_{ex}$ better characterized length scales relevant for microwave properties, we still compare our results to the

anisotropy derived from $p_c$. The length $p_c$ is defined by the slope at the origin of the correlation function and describes characteristics on the smallest length scales, e.g. the specific surface area (Löwe et al., 2011). The green dashed line in Fig. 6 shows the 5 cm running mean of the $p_c$-based anisotropy.

Naturally the anisotropy derived from $p_c$ deviates from the anisotropy derived from $p_{ex}$. This is plausible because a single correlation length cannot represent the complete snow microstructure. Especially for depth hoar, where the anisotropy derived

from $p_{ex}$ and $p_c$ differ most, the often used relation $p_{ex} \approx 0.75 p_c$ is not valid any more (Mätzler, 2002; Krol and Löwe, 2016). Indeed, we obtained for depth hoar layers a relation of $p_{ex} \approx 0.8...1.2 p_c$ (Fig. S18). Because $p_{ex}$ better describes microwave relevant length scales we focus in the following on comparisons with the $p_{ex}$-based anisotropy.

The comparison of simulated anisotropy profiles and CT-profiles shows that the model reproduces the CT-measured profiles in quite a detail (Fig. 6). Table 4(a) lists correlation coefficients between the simulated anisotropy and the 5 cm running mean

of the $p_{ex}$-based anisotropy (left columns) as well as correlation coefficients between simulated profiles and the individual CT anisotropy data points derived from $p_{ex}$ (right columns). For the 5 cm running mean, the Pearson-$r$ correlation coefficients are around 0.8 and higher except for CT-2 ($r = 0.54$) for which the snow structure does not show much vertical variability except for a thin layer of depth hoar at the bottom of the snow pack.

Despite of the good agreement of modeled and measured anisotropy data, a possible deficit of either the model, the definition

of the anisotropy, or the determination of the anisotropy by CT can be recognized for negative anisotropies $A < -0.1$. For example, for CT-1 in Fig. 6(a) at a height of 15–30 cm the model does not capture several layers that have undergone strong metamorphism. As shown on the right axis, these layers have manually been classified as depth hoar, DHcp/DHch, code according to Fierz et al. (2009). The CT data show clearly vertical structures with $A^{CT} \approx -0.15$. For these layers, depth hoar is also visible in a NIR image, Fig. 8(b), and also in the SNOWPACK grain classification, supplementary Fig. S19. Unfortunately,

no CT-data is available for the lowest 10 cm because the brittle structure of depth hoar could not be sampled.

Similar to the discrepancy in CT-1, the model does not reproduce the 3 cm thick layer of depth-hoar below a melt-crust at the bottom of the snow pack in CT-2 (Fig. 6(b); also visible in the NIR image, Fig. 8(c)). The same holds for the lowest 20 cm of CT-3 and -4, Figs. 6(c), (d) and the NIR images Figs. 8(d), (e). The lowest 40 cm of the snow pack in Fig. 6(c) have manually not yet been classified as depth hoar but still as faceted (rounded) grains (FCso, RGxf). Nevertheless, SNOWPACK classified

these grains as depth hoar.

For comparison of the depth-averaged radar data with the CT data, small ticks in Fig. 6, above the snow height line (HS), indicate various depth-averaged anisotropy values: the gray, green, blue, and red ticks are the values of the CT-measured ($p_{ex}$, $p_c$), simulated, and radar-measured anisotropy. The radar-measured anisotropy (red ticks) contains a small error range which corresponds to the standard deviation of the radar measurements. Numerical values are provided in Table 4(b). All values are

close to zero ($A^{...}_{avg} \approx 0.05 \pm 0.06$), except for CT-2 ($A^{...}_{avg} \approx 0.18 \pm 0.02$) which was sampled after intense snowfall and relatively





**Table 4.** (a) correlation coefficients between the anisotropy profiles calculated from $p_{ex}$ as presented in Fig. 6. The first three columns are the correlation with respect to individual anisotropy data points; the rightmost three columns are correlations with respect to the 5 cm running mean of CT-samples. Table (b) shows the depth-averaged anisotropy values from CT, model and radar data.

| (a) | correlation coefficients relative to CT data, $p_{ex}$ | | | | | |
|---|---|---|---|---|---|---|
| | CT single samples | | | CT: 5 cm running mean | | |
| profile | $r$ | NS | RMS | $r$ | NS | RMS |
| CT-1 | 0.70 | 0.15 | 0.125 | 0.79 | 0.12 | 0.111 |
| CT-2 | 0.37 | 0.10 | 0.148 | 0.54 | 0.20 | 0.147 |
| CT-3 | 0.86 | 0.67 | 0.116 | 0.95 | 0.78 | 0.091 |
| CT-4 | 0.89 | 0.61 | 0.141 | 0.90 | 0.67 | 0.123 |

| (b) | depth-averaged mean values | | | |
|---|---|---|---|---|
| profile | $A_{avg}^{CT,p_{ex}}$ | $A_{avg}^{CT,p_c}$ | $A_{avg}^{mod}$ | $A_{avg}^{CPD}$ |
| CT-1 | 0.03 | 0.11 | 0.05 | 0.06 |
| CT-2 | 0.17 | 0.16 | 0.19 | 0.17 |
| CT-3 | 0.04 | 0.11 | 0.08 | 0.04 |
| CT-4 | -0.02 | 0.06 | 0.05 | 0.06 |

moderate temperatures such that the effect of TGM was weaker compared to snow setting which resulted in a preferentially horizontal microstructure over the entire depth of the snow pack.

## 5.2 Distribution of fit parameters $\alpha_1$ and $\alpha_2$

The four winter seasons were characterized by quite different snow conditions which made it interesting to determine the free model parameters $\alpha_1$ and $\alpha_2$ independently for each season. Furthermore, to analyze their sensitivity with respect to slightly different snow conditions we determined the parameters for each ensemble member of the 300 best SNOWPACK simulations. Figure 7 shows a scatter plot of $\alpha_1$ and $\alpha_2$ plotted over the global (4-seasons) cost function (Appendix A4) which is indicated by contour lines. Gray dots represent the global solutions $\alpha_1, \alpha_2$ derived from the entire set of four seasons between 2009 and 2013. Colored dots show the solutions for each of the four winter seasons. Black dots with error bars are the mean and standard deviations of the five different sets of solutions; numerical values are listed in Table 5. All solutions are very close to the global solution except for the 3rd season (2011/2012). This season was characterized by a very fast transformation of 60 cm of fine grained, horizontally structured snow in Jan/Feb 2012 during very strong average temperature gradients of ∼65 K/m. The free parameters $\alpha_1$ and $\alpha_2$ of this season are about 1.5 to 2-times larger compared to the other seasons.





**Figure 7.** Solutions for $\alpha_1$ and $\alpha_2$ of all 300 best SNOWPACK runs for different seasons (colors) and all seasons combined (gray). Contour lines show the cost function, Eq. (A1), of all seasons combined. Table 5 lists mean and standard deviation of the set of solutions. Both values are visualized as dots and error bars above.



**Figure 8.** NIR Photography of the snow pack. The image NIR-0 was acquired in the first season on 2010-02-23 where no CT data is available. The other images NIR-1, -2, -3, -4 corresponds to the CT-profiles CT-1, CT-2, CT-3, CT-4. The intensity of the NIR photography is mainly determined by grain size but also shows nicely the metamorphic state of the snow pack. The NIR photographs provide an independent measure for the absolute depth of individual snow layers and help to identify strong structural transition in the snow pack.





**Table 5.** Parameter $\alpha_1$ and $\alpha_2$ estimated for different seasons. Provided are mean value and standard deviation.

| season | $\alpha_1$ | $\alpha_2$ |
|---|---|---|
| 2009 / 2010 | $0.57 \pm 0.04$ | $0.44 \pm 0.03$ |
| 2010 / 2011 | $0.64 \pm 0.03$ | $0.35 \pm 0.02$ |
| 2011 / 2012 | $0.98 \pm 0.13$ | $0.72 \pm 0.03$ |
| 2012 / 2013 | $0.68 \pm 0.07$ | $0.41 \pm 0.04$ |
| 2009–2013 | $0.69 \pm 0.04$ | $0.46 \pm 0.02$ |

## 6  Discussion

A main motivation of this paper was to show that it is possible to model the radar-measured anisotropy solely based on meteorological data. This was achieved in great detail and demonstrates that polarimetric radar measurements at sufficiently high frequencies (10–20 GHz) can be used to monitor the structural evolution of the snow pack nondestructively (Leinss et al., 2016) and even from space (Leinss et al., 2014).

Beyond that we provide a model which is able to reproduce the vertically resolved anisotropy that was derived from tomographic reconstruction of the snow pack. Furthermore, it is remarkable that a model, which completely neglects any microstructural parameters like grain size, SSA or snow classification is able to simulate the temporal evolution of a microstructural parameter, the anisotropy, solely based on macroscopic fields. Indeed, the model lacks several details which will be discussed in the following sections after providing a general overview of the model results.

### 6.1  Discussion of model results and snow conditions

This section discusses the evolution of the simulated and radar-measured anisotropy with respect to general weather and snow conditions. Snow conditions differed significantly between the different winter seasons, therefore we provide a short summary of snow conditions for every season. For reference, snow height, air temperature and soil temperature are plotted in Figs. 9 and 10.

In the first season, 2009/2010, snow fall started early October and accumulated up to 30 cm with relatively moderate temperatures (and some short melt events) until mid of December 2009 when temperatures dropped well below zero and caused soil freezing. Four major snow fall events followed until April when snow melt set it.

The simulated profile for the first season, 2009/2010, Fig. 4(a), shows a strongly varying anisotropy in Oct/Nov which transforms into vertical structures with the cold temperatures in early January. The following four major snow fall events appear as an increase of the average anisotropy in Fig. 4(b). As no CT-data are available for the first season, we provide a NIR image from 2010-02-23 in Fig. 8(a). The NIR image confirms the model results of metamorphic snow (depth hoar) in the lower 30 cm of the snow pack and shows multiple distinguishable layers above. In Oct/Nov 2009 model and radar measurements do not agree because the 20 cm thin snow pack results in a very imprecisely radar-measured anisotropy (green line vs. gray dots in



**Figure 9.** Snow height, air- and soil temperature at different locations (IOA, AWS, MetM = meteorological mast) for the first two seasons, 2009/2010 and 2010/2011.

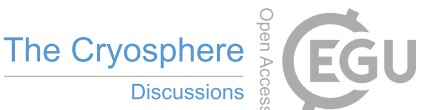



**Figure 10.** Snow height, air- and soil temperature at different locations (IOA, AWS, MetM = meteorological mast) for the last two seasons, 2011/2012 and 2012/2013.



Fig.4(b)). Additionally, because of limited microwaves penetration, short surface melt events appear as radar-anisotropy values around zero.

In the second season, 2010/2011, conditions are characterized by cold temperatures below -10° C already in early November combined with a shallow snow pack such that soil freezing occurred in the second week of November. Until January snow height was less than 30 cm and an over 20 cm thick layer of depth hoar was present during the entire season (NIR image, Fig. 8(b)).

The simulated profiles for the second season, 2010/2011, Fig. 4(c), shows that the snow from early November transformed to vertical structures within 2–3 weeks. These vertical structures persisted through the entire winter season. In the radar-signal in Fig. 4(d), the vertical structures appear clearly as a negative signal until January after which 35 cm of additional snow accumulated. The comparison with CT data from 2011-03-03, Fig. 6(a), shows a good agreement in the upper snow layers but a significant discrepancy between 10 and 30 cm where the model predicts positive values $A^{\mathrm{mod}} \approx 0.05$ but the CT data show negative values $A^{\mathrm{CT,p_{ex}}} \approx -0.15$.

In the third season, 2011/2012, snow fall started later as usual, but during December 2011 about 50 cm of snow accumulated during very mild air temperatures, often above -5°C. As settling dominated TGM, almost the entire snow pack showed a horizontal structure until mid January (Fig. 5(a) and CT-2 in Fig. 6(b)). Only a few-centimeter thick layer of depth hoar (below a melt-crust layer) was present which is, similar to the previous season, only poorly represented by the simulation. The mild temperatures in December caused very weak TGM which preserved the fine-grained snow, clearly visible in the NIR photo Fig. 8(c). Mid-January to early February, temperatures dropped gradually from -10°C to -30°C and strong TGM set in (compare Fig. 8(c) with Fig. 8(d)). Then, mid of February about 30 cm of fresh snow accumulated ontop of the strongly transformed lower layers which is visibly by the settling-induced horizontal structures ($A \approx +0.25$) in the upper 30 cm of the snow pack in the anisotropy profile CT-3, Fig. 6(c). After that, several minor snow fall events repeated until snow melt end of April which appeared as little oscillations in the radar-measured anisotropy, Fig. 5(b).

It is worth to note that for the period Jan/Feb 2012, both the simulation, Fig. 5(a), and the comparison of the CT-profiles (CT-2 and CT-3 in Fig. 6) show that the entire snow structure transformed from horizontal to vertical structures. However, the model was not able to track the fast decay of the anisotropy between January and February 2012 and generated too positive anisotropy values for the minimum mid of February ($A^{\mathrm{mod}}_{\mathrm{avg}} \approx 0$, Fig. 5(b)) for which a negative average anisotropy of $A^{\mathrm{CPD}}_{\mathrm{avg}} \approx -0.05$ was measured by radar. Field measurements and CT data show that in the early winter 2011/2012 finer grain sizes were observed compared to other winter seasons (Leppänen et al. (2015) and SSA data in Fig. S18). It is likely, that the fine grains transformed faster than represented by the model. The resulting offset of a systematically too positively modeled anisotropy persists from February until snow melt. Except for the offset, the subsequently simulated anisotropy variations agree well with the radar-measured anisotropy.

At the end of the third season, from 10–13th of April 2012, wet snow and rain fell ontop of the snow pack which froze afterwards. This causes an interesting feature at the end of the season in Fig. 5(b): the rain-on-snow event appears as a dip in the previously positive values of the radar-measured anisotropy. This event coincided with strong settling and the influence of the latter on the anisotropy was balanced by adjusting the melt metamorphism term, Eq. (14). In contrast to the dip in the radar





measurements, the simulated time series (green line in Fig. 5(b)) shows a positive peak in the average anisotropy indicating strong snow settling. This apparently opposite trend likely results from limited microwave penetration into the wet snow during the time when settling of the wet snow was strongest.

In the last season, 2012/2013, conditions are characterized by four major snow fall events and almost no precipitation from February until April 2013. During the period of the first snow fall events in November, temperature was frequently at 0°C such that no reliable radar measurements were possible (initial "noise" indicated by gray dots in Fig. 5(d)). The last dip in the radar data end of November 2012 (gray dots in Fig. 5(d)) is an indicator for increasingly wet snow. This goes along with positive temperatures (Fig. 10) and also simulated liquid precipitation (Fig. S13) which caused a height loss of about 10 cm due to rain on snow which was not correctly simulated by SNOWPACK (see also Fig. A6). As a consequence SNOWPACK modeled not enough precipitation mid of December; therefore, the peak in the mean modeled anisotropy (mid of Dec 2012 in Fig. 5(d)) which indicates horizontal structures is too small compared to the radar data. Then, starting from December 2012, the simulated anisotropy, Fig. 5(c), shows a persistent 30 cm thick layer of depth hoar which continued to evolve during the remaining season. This depth hoar layer formed after the melt event end of November after which air temperatures suddenly dropped to around -20°C. This depth hoar layer reached the lowest anisotropy values observed $A^{\mathrm{CT},p_{\mathrm{ex}}} \approx -0.3 \pm 0.1$ (CT-4 in Fig. 6(d)). Similar to the season 2010/2011, the model seems either to overestimate such low anisotropies or the CT-data underestimates the anisotropy of depth hoar layers. Nevertheless, the average anisotropy of the remaining season was well simulated, Fig. 5(d).

Interesting in March and April 2013 are the modeled vertical structures close to the snow surface which also appeared in other seasons. These are explained by strong snow surface temperature gradients which act on the snow at the surface while this snow does not experience any overburdened pressure and can therefore quickly transform into vertical structures or possibly surface hoar as classified by SNOWPACK (Figs. S19).

## 6.2 Discussion of meteorological input data

For best results of the anisotropy model it is critical that both, meteorological input data and snow properties simulated by SNOWPACK are as correct as possible. For most of the meteorological data this was ensured by redundant sensors as shown in Figs. 9 and 10 and discussed in Appendix A1 (for data of all sensors see Fig. S2–S5). Only precipitation and solar radiation required some calibration. Precipitation was adjusted using SWE measurements (Appendix A1). Still, some inaccuracies in precipitation data can be detected by comparing measured and modeled snow height, Fig. A5. Radiation data was homogenized and gaps were filled as described Appendix A2. The gaps can be found in Figs. S2–S5 or likewise in Fig. S14. Solar radiation was calibrated implicitly by calibration of SNOWPACK as described in Appendix A3. Unfortunately, snow temperature measurements, used for calibration, contained also some unrealistic values. In the following we discuss which gaps and errors in radiation, precipitation and snow temperature could have affecting the simulated anisotropy.

For snow temperatures, we noticed that in February 2011, when air temperatures dropped below -30°C, modeled snow temperatures were 10–20 K higher than measured snow temperatures (red vs. black lines, 2nd column in Figs. A5 and A6). We are quite confident that this is a measurement error because for the 60 cm thick mid-winter snow pack it is physically





unrealistic that the measured snow temperature at +10 cm above ground deviates strongly from the measured soil temperature 5 cm below ground. A similar effect might have occurred in Feb 2010 where measured snow temperatures +50 cm above ground were about 10 K colder than modeled temperatures. The reason could be a few cm deep snow pit at the sensor array as mentioned in Sect. 3.3. Fortunately, for both events, the modeled temperature at the bottom of the snow pack agrees very

closely to the measured soil temperature (red vs. gray line in the second-last row of Figs. A5 and A6). Hence, we are confident that SNOWPACK generally simulated quite reasonable snow temperatures.

For the long wave radiation data in the first season, several multiple day long gaps between Dec 2009 and Jan 2010 were interpolated and the results seems to be fine: modeled and measured snow height agree within a few cm when snow height was not enforced (Fig. A5) and SWE agreed within 10 mm when snow height was enforced (Fig. A6). Nevertheless, too high

incoming long wave radiation in the first week of January 2010, resulting in modeling of a too warm snow surface, could explain why the anisotropy in January 2010 did not decrease as indicated by the radar measurements, Fig. 4(b).

In the second season, several gaps of multiple days in the long wave radiation data between Nov 2010 and Jan 2011 seem to be correctly interpolated as both, snow height and SWE agree very well; for this period, simulated snow temperatures look reasonable. Also, the simulated anisotropy looks reasonable and agrees well with the radar data. In the third seasons, data was

complete and missing radiation data in Oct 2011 can be ignored because of snow free conditions.

In the forth season, the rain on snow event, mentioned in the previous section for late Nov 2012, was not correctly modeled by SNOWPACK. Although snow height was enforced, apparently no snow height loss was simulated (Fig. A6). This, possibly because of a gap in the radiation data where the actual incoming long wave radiation was likely higher than interpolated for the simulation. Simulated snow temperatures well below zero (Nov 2012, last column of Fig. A6) support the hypothesis that

incoming long wave radiation was filled with too low values. Because snow height was enforced, the too large snow height end of Nov implied forcing with too low precipitation for mid of Dec which resulted in less fresh snow with a positive anisotropy and in turn explains why the simulated anisotropy is lower than the radar-measured anisotropy, Fig. 5(d).

Missing short-wave reflection data were no problem, because short wave reflection was estimated based on the the simulated albedo. The incoming short wave radiation data did not contain any significant gaps.

## 6.3 Model deficits

In the model, we neglected any microstructure and instead introduced free parameters which were determined by the radar data. As single parameters cannot represent the underlying dynamics of the microstructure we discuss here the free parameters and neglected microstructural effects.

The scatter of the free model parameters $\alpha_1$ and $\alpha_2$ provides an uncertainty range and characterizes how specific these

parameters are for each season. In general, their values are close together, except for the set of parameters which provides the best solution for the third season (blue dots in Fig. 7). For this season, the values are almost twice as large as for the other seasons. Fig. A4 provides anisotropy profiles of all four seasons but which were calculated for $\alpha_1 = 0.98$ and $\alpha_2 = 0.72$ as determined for the third season. For this realization of the model, the mean-anisotropy shows stronger short-term variations, similar to overshooting, compared to the anisotropy measured by radar. Nevertheless, for this configuration the simulation



results seem to agree better with the CT-data, especially for strongly metamorphic snow. We conclude from that, that the range of $\alpha$-values provided in Table 5 reflects a similar uncertainty as the uncertainty for the anisotropy profiles calculated from CT data (see next section).

It is remarkable how well the model reproduces the radar-measured anisotropy time series. Nevertheless, it may surprise that the model completely neglects any dependence on grain size. However, we found that no simple grain size dependence, like weighting the TGM-term by the inverse grain size (by setting the microstructural parameter equal to grain size, $f_\mu = r_g$ in Eq. 9), could produce reasonable simulation results. Using the relation $f_\mu = r_g$ caused a strong vertical variability of the anisotropy combined with too positive values for the anisotropy of depth hoar (Fig. S24). We still think that neglecting the microstructure could be the main reason why the model was not able to simulate the fast decay of horizontal structures in Jan–Feb 2012.

Beyond the dimensions of the microstructure, we ignored the crystallographic fabric of snow, i.e. the orientation of the c-axis of the hexagonal ice crystals which compose the microstructure. For the radar data it was ignored because the snow fabric anisotropy affects only very weakly the dielectric anisotropy (Appendix A in Leinss et al. (2016)). For the model, we neither consider the evolution of the snow fabric anisotropy nor the influence of crystal orientation on the evolution on the structural anisotropy. This, because only very few studies exist which provide experimental insight about the orientation of the snow fabric (Calonne et al., 2016) or even the temporal evolution of the snow fabric anisotropy (Riche et al., 2013). Furthermore, the dominant growth direction of snow crystals depends on temperature (Lamb and Hobbs, 1971; Lamb and Scott, 1972) and is not necessarily parallel to the temperature gradient (Miller and Adams, 2009) as it can be clearly observed in the supplementary movie in (Pinzer et al., 2012). Motivated by the competing effect of crystal orientation, structural disorder and structural optimization to increase the vertical thermal conductivity (Staron et al., 2014) we simply introduced a lower limit of the anisotropy $A_{min}$ under TGM.

### 6.4 Characterization of the microstructure

As observed in Sect. 5.1 and discussed in Sect. 6.1, the largest deviations between the modeled and CT-measured anisotropy were found for depth hoar, where the modeled anisotropy is less negative than suggested by the CT-measurements based on $p_{ex}$. Interestingly, the model agrees well with anisotropy derived from $p_c$. However, we believe that the better agreement between the model and the anisotropy from $p_c$ compared to $p_{ex}$ is a coincidence which is not of further benefit for the interpretation of the radar measurements. This is supported by Fig. A4 where the parameters $\alpha_1$ and $\alpha_2$ determined for the third seasons 2011/2012 have been applied to all four winter seasons and where the simulated anisotropy is closer to the anisotropy derived from $p_{ex}$.

The origin of this discrepancy for depth hoar is not clear, especially in the context that for the second season the radar time series and the simulation, Fig. 4(d), agree best compared to other seasons (Pearson-$r > 0.97$, Table 3) while for this season the correlation with CT-data is the lowest (Table 4). We think that a combination of the following factors might have lead to the observed discrepancy: 1) uncertainties of the model to describe the dynamics of the anisotropy for depth hoar, 2) uncertainties in the CT-measurements where, especially for large depth hoar crystals, sample size effects could lead to biased estimates of





correlation lengths, here of $p_{\mathrm{ex}}$, 3) uncertainties how well the anisotropy derived from the microwave permittivity can actually be compared to the anisotropy determined from $p_{\mathrm{ex}}$, and 4) uncertainties from depth-averaging of the model data and the assumption of a homogeneous (depth-averaged) density for the radar-determined anisotropy.

Because the model was written to explain the microwave measurements it could be biased towards the anisotropy derived
from radar. On one hand the assumption of an exponential correlation functions seems to be more applicable to depth hoar than other snow types (Krol and Löwe, 2016) but on the other hand, we expect in general higher uncertainties of estimated $p_{\mathrm{ex}}$ for large structures (depth hoar) because of larger statistical fluctuations of the correlation function in finite samples. A systematic analysis of the representative elementary volume for the two point correlation function is however presently missing. Therefore it remains elusive which of the mentioned factors is the most important one. Further studies on sample size effects and combined
radar and CT measurements could elucidate this problem.

On physical grounds, it is also reasonable to expect that $p_{\mathrm{ex}}$ rather than $p_{\mathrm{c}}$ is better suited to characterize the structural anisotropy in the dielectric tensor because $p_{\mathrm{ex}}$ characterizes the structure on length scales which are (still small but) closer to the wavelength of the radar. In contrast, density fluctuations on the smallest scales (namely those characterized by $p_{\mathrm{c}}$) solely characterize the ice-air interfaces and the anisotropy calculated in (Löwe et al., 2011) from $p_{\mathrm{c}}$ seems to be irrelevant for the
radar wavelengths. To understand this we recall that the anisotropy of the dielectric tensor is characterized by a second rank tensor that is computed by an integral over the anisotropic correlation function of the material (Rechtsman and Torquato, 2008). Under the assumption that the correlation function has an ellipsoidal symmetry, this integral can be evaluated exactly to express the anisotropy via ratios of exponential correlation lengths in different directions when computing the parameter $Q$ that determines the eigenvalues of the structural fabric tensor. This has been done in (Leinss et al., 2016, Appendix C) for
the dielectric permittivity and in (Löwe et al., 2013) for the thermal conductivity leading to an very good agreement with simulations. It must be noted though, that the validity of the assumption of an ellipsoidal symmetry of the correlation function was never investigated in detail. In addition, the mere characterization of snow solely by three (exponential) correlation lengths is also an approximation (Krol and Löwe, 2016) which renders an estimation of (likely existing) uncertainties of CT-based $p_{\mathrm{ex}}$ estimates rather difficult. Here simulations of computer-generated two-phase media with prescribed correlation and anisotropy
structure might be a remedy (Tan et al., 2016).

## 7    Conclusions

In this paper, a model for the temporal evolution of the structural anisotropy of snow was designed. The model is based on simple rate equations and requires solely the folowwing macroscopic fields: strain rate, temperature and temperature gradient of the snow pack, ideally depth-resolved, as input variables. These variables are provided by most of the more advanced
snow pack models, here we used SNOWPACK. In the model, the evolution of the anisotropy is driven by the following three contributions: snow settling leads to a preferentially horizontal structure, temperature gradient metamorphism causes growth of vertical structures and melt metamorphism causes rounding of the structures. The three contributions are balanced by free




parameters which were calibrated by minimizing the difference between the modeled anisotropy and anisotropy data obtained from polarimetric radar measurements during four winter seasons between 2009 and 2013.

The results of the model, four years of depth-resolved anisotropy time series, were validated with computer tomographic (CT) measurements of the snow microstructure acquired during four field campaigns. For validation, depth-resolved anisotropy
profiles were determined from the CT data via exponential correlation lengths $p_{ex}$ derived from two-point correlation functions.

The model results are remarkable in several aspects: First the detailed agreement between radar-measured anisotropy and the anisotropy modeled solely based on meteorological input data demonstrates that polarimetric radar measurements at sufficiently high frequency (10–20 GHz) can be used to monitor the structural evolution of the snow pack. Second, the good agreement of the model with CT measurements shows that a microstructural parameter like the anisotropy can be modeled solely based
on macroscopic fields. Third, the model demonstrates that it can reproduce the anisotropy which can only be determined with sophisticated CT or radar measurements. Nevertheless, for depth hoar we found that the model significantly underestimates the strong vertical structures which result from CT-data when deriving the anisotropy from $p_{ex}$.

Finally, the large observation time spanning four winter seasons with a sampling interval of four hours builds an unique data source to study the evolution of the anisotropy of snow. We think, that the developed model and the determined parameters
are relevant for future consideration of the anisotropy in snow models. Beyond that, the well calibrated SNOWPACK model provides an additional data set to study microwave properties of snow especially within the framework of the Nordic Snow and Radar Experiment (NoSREx-I-III) in Finland, Sodankylä.

With the long time series and the developed model we gained a deeper insight into the growth mechanisms of anisotropic snow crystal and identified the two main driving terms, the strain rate and the vertical water vapor flux. The model could
help to enhances the understanding of macroscopic anisotropic properties like thermal conductivity, mechanical stability and electromagnetic properties, and demonstrates that the anisotropy can be measured by means of polarimetric radar systems which provides a new method to access microstructural properties of snow non-destructive and even from space.

## 8 Data availability

All data are originally from the NoSREx-campaigns (Lemmetyinen et al., 2016, 2013) and are partially available at www.
litdb.fi. Radar data are available from ESA or the GAMMA Remote Sensing and Consulting AG. Preprocessed meteorological input data, configuration files and simulated snow profiles from SNOWPACK, modeled anisotropy time series, radar-measured anisotropy time series, SWE measurements and CT-data are available under DOI:http://dx.doi.org/10.3929/ethz-b-000334041.

## Appendix A: Preprocessing of meteorological data and SNOWPACK calibration

### A1  Preprocessing of meteorological data

In order to provide SNOWPACK physically consistent input data all meteorological data were preprocessed, filtered, combined and gaps were interpolated if they could not be filled by data sets of equivalent sensors. Figure S1 shows a processing flow chart





of the meteorological data which was used to create the three input files required by SNOWPACK (soillayer*.sno, config*.ini, meteoin*.smet). We combined data measured at the IOA (MAWS\*), meteorological mast (arcmast\*), and from the AWS. All raw data were downsampled to a 1 hour sampling interval. Invalid data were removed and equivalent datasets were averaged. Data gaps were interpolated with algorithms which considered diurnal and seasonal cycles and also the type and statistics of

existing data series (details below).

Snow height (HS) and air temperature (TA) were measured by at least one sensor at each of the three site (IOA, AWS, meteorological mast), but some of the data series contained gaps for periods of a few days. The measurements of the three sensors were very similar (see supplementary figures S2–S5; standard deviation snow height $\sigma_{HS} = 2.6\,\mathrm{cm}$, max. difference $\Delta HS_{95\%} < 10\,\mathrm{cm}$ for 95% of measurements. Standard deviation of air temperature $\sigma_{TA} < 0.6\,\mathrm{K}$, max deviation of air tempera-

ture $\Delta T_{95\%} < 2.0\,\mathrm{K}$ for 95% of measurements.). Therefore the data were averaged when data from more than one sensor were available. By this redundancy, we obtained almost complete time series of snow depth and air temperature. Remaining gaps of a few days were interpolated.

Four different soil temperature measurements (TSG) were averaged: they were measured at each two locations 2 cm below the surface few meters apart at the IOA (SMT: soil temp B, soil temp C) and at two sites near the meteorological mast at -5 cm

and -10 cm depth. The soil temperature of all four sensors differed less than 1.5 K for 95% of measurements and had a standard deviation of 0.5 K (see supplementary figures S2–S5).

Soil moisture showed signification variations between the six different sensors (each two sensors at -2 cm and -10 cm depth at the two locations SMT-A and SMT-B at IOA and also two sensors at -5 cm and -10 cm depth at the meteorological mast). However, all sensors showed the same trends with 5–15% $_{\mathrm{vol}}$ liquid water content during summer, 1–3% $_{\mathrm{vol}}$ liquid water content

during winter and 15–35% $_{\mathrm{vol}}$ liquid water content during snow melt (supplementary figure S2–S5).

Relative humidity (RH), wind speed (VW), wind direction(DW), and maximum wind speed (VWM) was only measured at the AWS and gaps of a few days were filled by a combination of linear interpolation, average data from the four seasons and diurnal cycles.

Precipitation (PSUM) was measured 600 m north of the IOA. In order to calibrate the precipitation data to the IOA, we

adjusted the precipitation data such that the cumulated precipitation of the AWS ($SWE_{AWS,cal}$) follows closely the reference snow water equivalent ($SWE_{REF}$), composed by measured SWE data of the SSI and the GWI. Calibration was done by amplifying/decreasing existing precipitation when the cumulated precipitation of the AWS, $SWE_{AWS,raw}$, was lower/higher than $SWE_{REF}$. A comparison of raw precipitation ($P_{AWS}$, blue), calibrated precipitation ($P_{REF}$, red) and precipitation change (green) are shown at the top together with the SWE data (below) in Fig. A1. SNOWPACK runs with calibrated and uncalibrated

precipitation showed that the calibration of precipitation improved the results for the simulated snow height.

The precipitation phase (PSUM_PH) was measured by the distrometer located at the IOA (data from www.litdb.fmi.fi). However, the data was not directly used because the distrometer frequently misclassified snow as rain. Therefore, the distrometer data was only used to check the rain/snow threshold (THRESH_RAIN). According to the distrometer data the rain/snow threshold is at $T = 0.73°C$ or alternatively a linear range from $T_{snow} = 0.06°C$ to $T_{rain} = 1.40°C$ was obtained (Fig. A2).



## A2  calibration and interpolation of radiation data

To provide consistent solar radiation data, data acquired by different sensors between January 2009 and September 2015 were homogenized and gaps with missing data were interpolated. Plots of the original raw data and the homogenized and filled data are shown in the supplementary material (Supplements, Fig. S14).

Short wave radiation was measured at the sounding station. The short wave sensors were replaced in August 2012, therefore the radiation time series were homogenized to provide consistent time series. Homogenization was done by increasing the reflected radiation by 31% such that the summer albedo was similar before and after sensor replacement. Furthermore, reflected short wave radiation which exceeded incoming short wave radiation was set to 95% of the incoming short wave intensity. Values with reflected short wave radiation of zero were considered as invalid data when the incoming radiation exceeded 20 W·m$^{-2}$.

Between 10 November 2011 and 23 March 2012 the reflected short wave sensor malfunctioned. The gap was filled by the product of the incoming short wave radiation and the albedo averaged for every day of the year over the period 2009–2015. The albedo at the onset of snow fall (09–19 Nov 2011) was interpolated by manual estimates. Remaining short data gaps in the reflected short wave radiation data of a few hours were interpolated via the albedo on which an Gaussian average of neighboring pixels was applied (FWHM = 1 day, kernel size = 12 days).

The long wave radiation balance was measured at the radiation tower. Long wave radiation data contained a few gaps up to 20 days long (one gap of 52 days in autumn 2011 is irrelevant because this gap is before the onset of snow fall). Data gaps shorter than 12 days were interpolated by the Gaussian average (FWHM = 1 day, kernel size = 12 days) of neighboring data points. Remaining gaps of up to 8 days were linearly interpolated. Additionally, to reconstruct the diurnal radiation cycles, the average radiation of each hour of the year was high-pass filtered (Gaussian window of 6 days) and added to the smoothly
interpolated data gaps (for plots of the raw and interpolated data see supplementary material, Figs. S2–S5, S6–S9, and S14).

## A3  SNOWPACK calibration

For comparison of the natural snow pack with the modeled snow pack under different configuration settings, we compared measured and modeled snow height and snow temperature. Snow temperature was measured at five internal snow temperatures sensors at 10, 20, 30, 40, and 50 cm above ground. For snow height and snow temperature we evaluated for each of the four
season each six statistical descriptors: the smallest (negative) difference, the largest (positive ) difference, the absolute deviation for which 95% of all absolute deviations are smaller, the root mean square error, the mean difference, and the Nash-Sutcliffe model coefficient. Additionally, we calculated these descriptors for all four seasons together. This provided in total 60 quantities for comparisons. To determine the "best" simulation(s), we compared this 60 quantities of every SNOWPACK run with all of the other 5000+ SNOWPACK runs and calculated a score which describes how many times these 60 comparisons show a better
result (smaller error, larger Nash-Sutcliffe coefficient) than all other runs. The total score was divided by the total number of runs which results in a score between 60 and 0. A score of 60 indicates that a single run outperfoms every other run independent on which statistical variable is analyzed. The maximum achieved score was 51.3, the lowest score 9.3.





**Table A1.** Thresholds for snow height (HS) and snow temperature (TS) which were used to score the different SNOWPACK runs.

| statistical descriptor | | threshold value for | |
|---|---|---|---|
| evaluated for all/each year(s) | | HS (cm) | TS (°C) |
| smallest negative Difference | > | -17.5 | -9.5 |
| largest positive Difference | < | 17.5 | 14.5 |
| max. abs. difference (95%) | < | 10.5 | 3.4 |
| root mean square error | < | 5.3 | 1.80 |
| mean difference | < | 1.6 | 0.29 |
| Nash-Scliffe coefficient | > | 0.95 | 0.82 |

Additionally to the relative scoring by pair-wise comparison of all SNOWPACK runs, we used a second scoring scheme which defined height and temperature thresholds for each of the six statistical descriptors. The thresholds are listed in Table A1. The sum of all fulfilled conditions for all years simultaneously and for all individual years made again a maximum score of 60. The score by comparison and score by threshold show an approximately linear relation. Histograms over all SNOWPACK

runs with the score by threshold, and the distribution of statistical descriptors are shown in Fig. S15

For SNOWPACK calibration, we varied the following parameters: scaling of short wave and long wave radiation by various constant factors, various thresholds for the snow/rain threshold (THRESH_RAIN), various factors for the WIND_SCALING_FACTOR with SNOW_EROSION = TRUE/FALSE, five different settings for the ATMOSPHERIC_STABILITY, creation of short wave reflected radiation from albedo (RSWR::create = ISWR_ALBEDO) on/off, Calibrated or uncalibrated precipitation PSUM

(see section A1), with or without provided precipitation phase (PSUM_PH in *.smet files), filling of long wave radiation gaps with the generator ILWR::allsky_lw::type = Konzelmann or our method described in section A2, and SW_MODE = BOTH/INCOMING.

We found, that radiation scaling was crucial to produce correct results. Additionally snow erosion with a wind scaling factor around two significantly improved the results. Only with atmospheric stability = normal, we got good results but the

other stability models were also not too far from reality. Interestingly, only the model MO_MICHLMAYR required not much modification of the radiation in contrast to the other atmospheric models. Setting SW_MODE = INCOMING instead of BOTH did not change the results except near the end of snow melt where a slight change was observable. Obviously, for our test site, SNOWPACK works better when the reflected short wave radiation is estimated via the albedo than vice-versa.

## A4  Cost function for determination of $\alpha_1$ and $\alpha_2$

For iterative determination of the free model parameters $\alpha_1$ and $\alpha_2$ we defined a cost function which describes approximately the root mean square error between the depth-averaged simulated anisotropy, $A_{\mathrm{avg}}^{\mathrm{mod}}$, and the radar measured anisotropy, $A_{\mathrm{avg}}^{\mathrm{CPD}}$. The cost function is evaluated only for the set $M$ of radar measurements acquired during dry snow conditions. With this set $M$

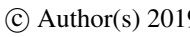



of about 3200 measurements, where each measurement has the index $i$, i.e. $i \in M$, we defined the cost function $J(\alpha_1, \alpha_2)$ as

$$J(\alpha_1, \alpha_2) = \left[ \frac{1}{N} \sum_{i \in M} \left( A_{\mathrm{avg},i}^{\mathrm{mod}} - A_{\mathrm{avg},i}^{\mathrm{CPD}} \right)^2 \right]^{1/2}$$

$$+ \left[ \frac{1}{N} \sum_{i \in M} \left( A_{\mathrm{avg},i}^{\mathrm{mod}} - A_{\mathrm{avg},i}^{\mathrm{CPD}} - \overline{\Delta A}_{i|i \in M_y} \right)^2 \right]^{1/2}. \tag{A1}$$

In the second term, the subset $M_y$ of $M$ (with $M = \cup_{y=1}^4 M_y$) defines all radar measurements acquired during an individual

winter season $y$ with $y = 1, 2, 3, 4$. In this term, we subtract for each season the mean difference between modeled and measured data. This mean difference, defined by $\overline{\Delta A}_{i|i \in M_y} = \langle A_{\mathrm{avg},i}^{\mathrm{mod}} - A_{\mathrm{avg},i}^{\mathrm{CPD}} \rangle$, where $\langle \cdot \rangle$ indicates the mean, regularizes the cost function as described below.

The regularization is required because the first term in Eq. (A1), evaluated for the range of $\alpha_1$, and $\alpha_2$, is characterized by an elongated valley structure indicted by contour lines in Fig. A3(a). In this valley, the shallow global minimum is indicated

by a white cross. To avoid ambiguities in the shallow minimum of the cost function we added the second term in Eq. (A1) which evaluation shows a substantially different structure, Fig. A3(b), than the first term. The different structures originate from the regularization using the mean difference between $A_{\mathrm{avg}}^{\mathrm{mod}}$ and $A_{\mathrm{avg}}^{\mathrm{CPD}}$. This regularization acts as a high pass filter and allows for quantitative consideration of the simulated amplitude of the time-varying anisotropy time series without screening good solutions by seasonally constant (or linearly drifting) offsets. Such offsets can be caused when the anisotropy of the early

winter snow pack is not correctly modeled and persists through the entire winter season even though the anisotropy of younger layers accumulated ontop of the early winter layers is correctly modeled. For example, in season 2011/2012, Fig. 5b, the model does not catch the fast growth of vertical structures in Jan/Feb 2012 and an offset exists after February which persists until snow melt. Nevertheless, the amplitude of anisotropy variations between February and April 2012 are quantitatively correctly modeled.

The error space of the full cost function, comprising both terms of Eq. (A1), is shown in Fig. A3(c). The exploration of the cost function shows that the global minimum can be found with a local hill-climbing optimization method. Because computation of the cost function requires evaluation of all simulated snow layers and time steps a derivation-free algorithm significantly accelerates the optimization. We used the downhill-simplex method (Nelder-Mead or amoeba method). Additionally to the cost function, the model accuracy was measured with the Nash-Sutcliffe model efficiency coefficient which is shown in Fig. A3(d).

*Author contributions.* SL and HL wrote the manuscript, SL processed all meteorological and radar data and designed the model, HL processed all CT data, MP, HL and AK collected the field data.

*Competing interests.* The authors declare that they have no conflict of interest.





*Acknowledgements.* The *in situ* data collection was supported by the European Space Agency activity "Technical assistance for the deployment of an X- to Ku-band scatterometer during the NoSREx campaigns" (ESA ESTEC Contract no. 22671/09/NL/JA/ef) (Lemmetyinen et al., 2013). The staff at FMI-ARC is acknowledged for the collection of in situ data. Andreas Wiesmann from GAMMA Remote Sensing is acknowledged for technical assistance with the SnowScat data and Margret Matzl for the lab sampling procedures for casted DEP samples.

5 Special thank goes to Jouni Pulliainen from FMI for the initiative of setting up a test site which provides a unique amount and diversity of meteorological data and snow measurements. We thank Matthias Bavey from SLF for helping to find the best SNOWPACK configuration. Juha Lemmetyinen deserve a major thank for his support concerting all details about the test site and the field data. The paper was funded by ETH Zürich. Irena Hajnsek deserves a major thank for her patience and for the freedom to write this paper.



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





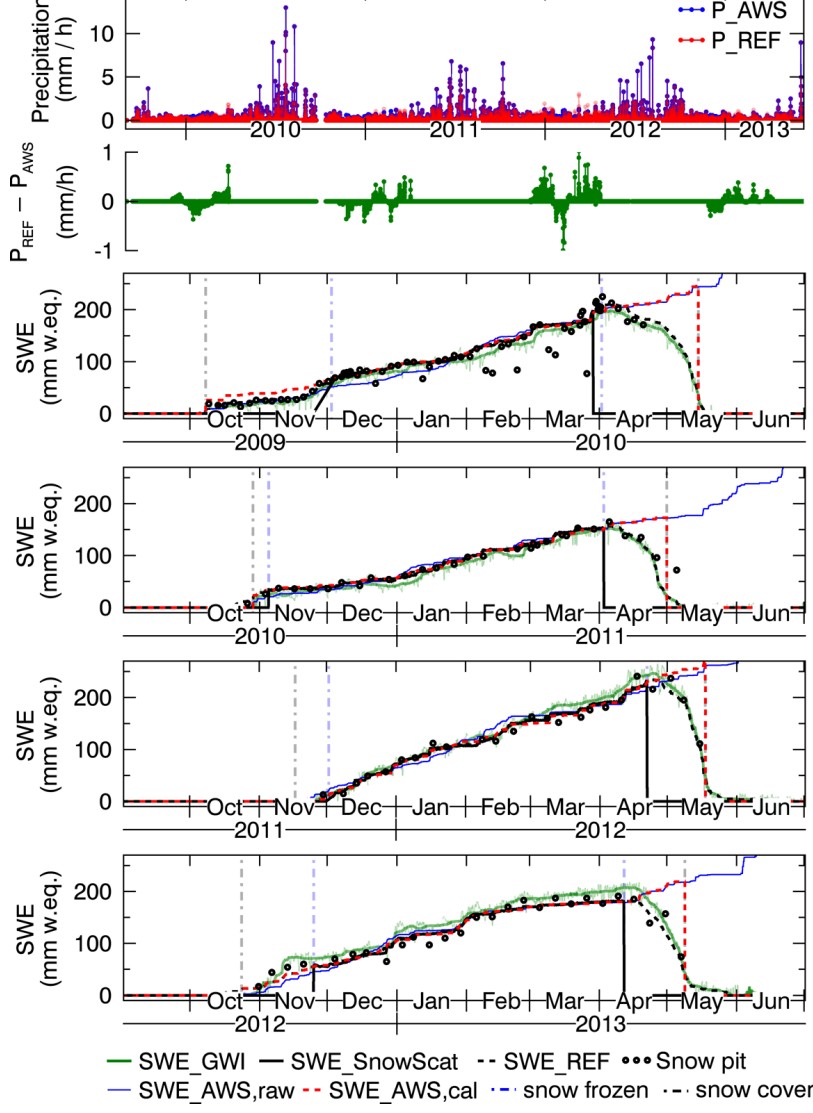

**Figure A1.** Precipitation and SWE data. Top: precipitation from the AWS and adjusted precipitation ($P_{REF}$) used to run SNOWPACK. Below in green: Difference between the original and adjusted precipitation data. The four SWE time series below were measured in the snowpit (black bullets), by the GWI (green), and by SnowScat (black). Blue and red lines are the cumulated precipitation of the AWS and the adjusted precipitation $P_{REF}$. Vertical dash-dotted lines indicate the time of snow freeze and melt (light blue) and the period of snow covered ground (gray). Snow water equivalent (SWE) was measured by the SnowScat instrument during dry snow conditions. SWE during snow melt was determined by the Gamma Water Instrument (GWI) which was calibrated by manual SWE measurements from the snow pit. To obtain complete time series of SWE (= $SWE_{REF}$) we composed the SWE signal from GWI measurements ($SWE_{GWI}$) during wet snow conditions and used radar measurements ($SWE_{SnowScat}$) during dry snow conditions. Details about the SWE measurements with SnowScat, the AWS and the GWI are published in (Leinss et al., 2015).



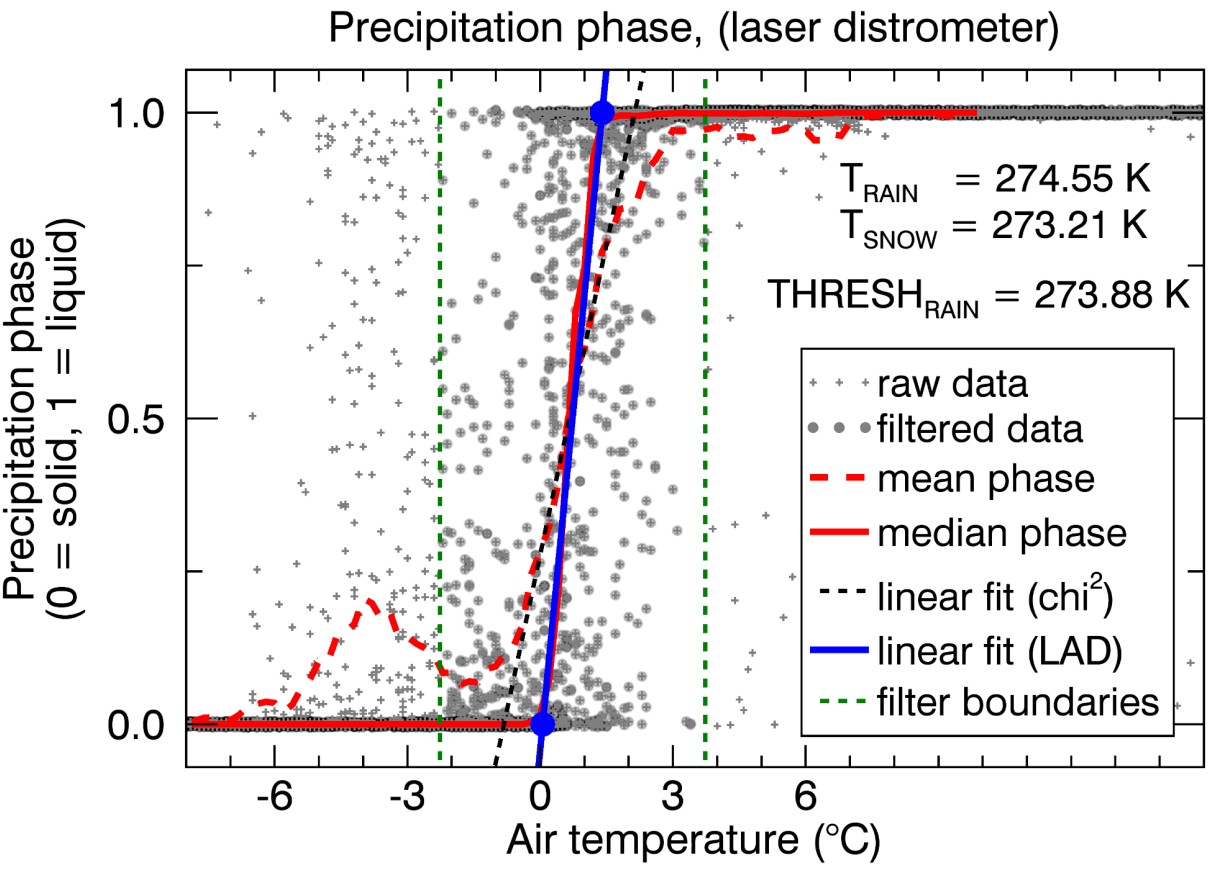

**Figure A2.** The precipitation phase was measured by the distrometer. From the precipitation phase and the air temperature measured by the AWS we determined a mean rain/snow threshold of 0.73°C using a robust least-absolute-deviation (LAD) fit to the data (blue line). A linear fit provides the same threshold but a slightly lower slope. Before fitting, we set a filter boundary (green dotted line) of 0.73±3°C. Data outside the filter boundary are considered as misclassified precipitation.



**Figure A3.** Cost functions for the difference between modeled and radar-measured anisotropy used to determine the free model parameters $\alpha_1$ and $\alpha_2$. Shown are the cost-functions for the best SNOWPACK simulation and the entire time range (2009–2013). The cost functions (a): root mean square error (RMS). (b): root mean square error with mean-difference subtracted (RMSc), (c): sum of RMS and RMSc, (d): Nash-Suttcliffe model efficiency coefficient. Cost-functions for individual seasons are provided in the Supplements.





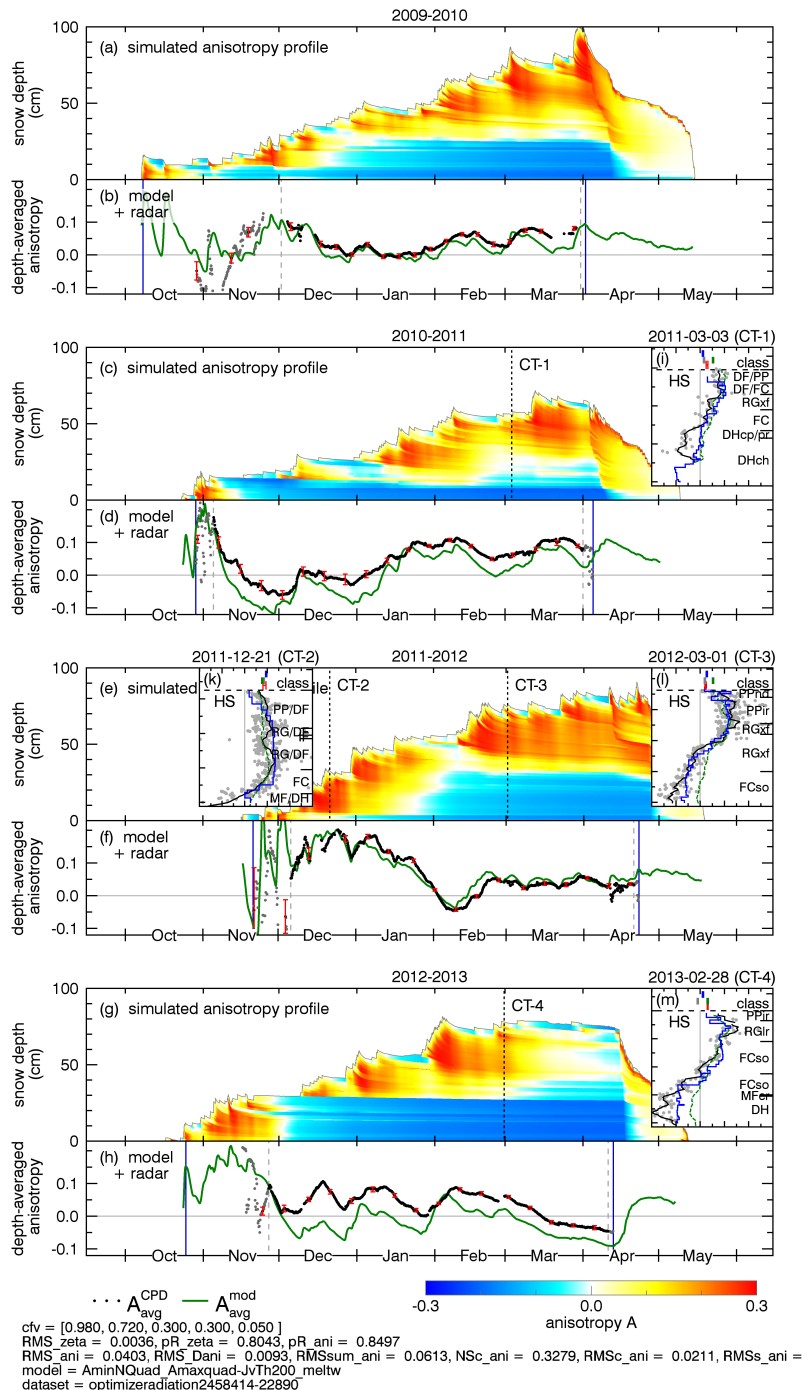

**Figure A4.** Modeled solution of all four seasons when the parameters $\alpha_1$ and $\alpha_2$ are optimized for the third season ($\alpha_1 = 0.98$ and $\alpha_2 = 0.72$).





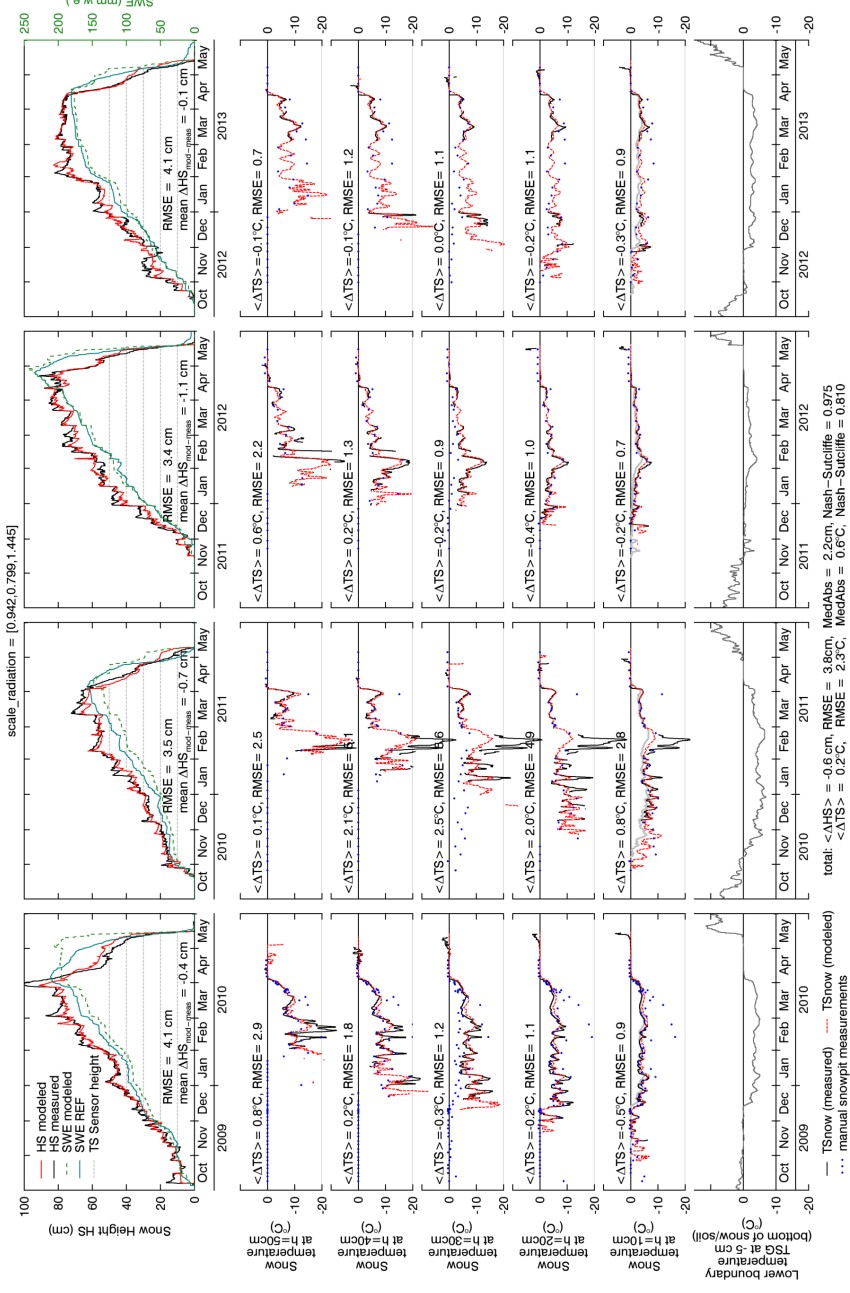

**Figure A5.** Measured and modeled snow height and SWE (top), and snow temperature at various depths (bottom) for the best SNOWPACK run where snow height was not enforced (ID: optimizeradiation2458414-31824). The gray line in the second-lowest row is equivalent to the -5 cm soil temperature (TSG, last row) and is helpful to identify unrealistically low measured snow temperatures (e.g. Feb 2011 and possibly Feb 2010 which can be identified very likely as measurement errors). For comparison, blue dots indicate manual snow temperature measurements in a snow pit.





**Figure A6.** Measured and modeled snow height, SWE, and snow temperature at various depths for the best SNOWPACK run where snow height was enforced (ID: optimizerradiation2458414-22890). The gray line in the second-lowest row is equivalent to soil temperature (TSG, last row) and can be used to identify unrealistically low measured snow temperatures (e.g. Feb 2011 and possibly Feb 2010 which can be identified very likely as measurement errors). For comparison, blue dots indicate manual snow temperature measurements in a snow pit.