# Peer review of "Modeling the Evolution of the Structural Anisotropy of Snow"

_The Cryosphere, 2019_

## Referee Comment (RC1) · Anonymous Referee #1 · 21 Jun 2019

**\*\* General comments**

Snow sometimes presents a structural anisotropy which induces an anisotropy in the physical and mechanical properties of snow. Although its influence can be significant, anisotropy is currently not taking into account in snowpack models. Contributing to address this gap, the paper presents a macroscopic model of the structural anisotropy of snow.
The model relies on physical concepts partly supported by observations from literature (field studies and cold-lab experiments). However, these observations are limited, so the physical concepts of the model also rely on some hypothesis. The model is empirical and includes free parameters that are calibrated from measurements.
Briefly, the model consists in empirical descriptions of three processes that influence structural anisotropy: temperature gradient metamorphism, settlement, and wet snow metamorphism. These three contributions are balanced by empirical parameters that are calibrated based on a large dataset of bulk structural anisotropy (average over the full-depth of the snowpack) obtained from radar measurements performed at Sodankyla, Finland, during 4 winter seasons, on dry snowpacks. The inputs of the model are: layer temperature, temperature gradient through the layer, strain applied on the layer, and liquid water volume fraction of the layer. There is one initial condition: all new snow layers have the same structural anisotropy that corresponds to a slight anisotropy towards horizontal structures.
Concerning evaluation, the model is assessed based on 4 vertical profiles of structural anisotropy at Sodankyla, Finland, at different dates between 2010 and 2014. To do so, they implemented the anisotropy model in the detailed snowpack model "SNOWPACK", run this version of the SNOWPACK model at Sodankyla with an adequate, tuned set of forcing, and compare the simulated profiles of anisotropy to the ones derived from computer-tomography measurements (Fig. 6 and Table 4 of the paper). They conclude on an overall good agreement, except for pronounced vertical structures that are significantly underestimated by the anisotropy model.

The paper addresses a very interesting topic with potential great outcomes for the snow science community. It is well written, although it would gain clarity by being shorten or more to the point in some paragraphs. The topic is clearly suitable for a TC publication. The studies is based on a large variety of methods (field measurements, simulations, analytical work). The work is highly novel since it is an attempt to the first model of structural anisotropy.
I have however some major comments that concern:
1/ The evaluation of the model is very limited, which do not allow to be really convinced about the presented work; yet, there are options to perform a more robust, more "instructive" evaluation.
2/ The physical processes involved in the anisotropy model are, for some of them, justified based on incorrect references (wrong interpretations or over-interpretation of previous studies) and more care should be given to presenting results from previous related studies. The distinction between what is based on assumptions and on observations should appear more clearly.
Besides, several unsupported comments can be found throughout the paper, that should be re-written (in minor comments).

I would like to strongly encourage the authors to address these main issues, as, I believe, with an improved evaluation and some efforts to better describe and justify the model design, the present paper would be very beneficial for the snow community.

**\*\* Specific comments (major)**

**1/ Robust evaluation is missing**
The validation step of the anisotropy model is too little addressed in the paper. The evolution of structural anisotropy has been only observed in some cases, and is not fully understood yet,

especially the evolution during settlement and wet snow metamorphism. The proposed model relies thus partly on hypothetical processes. It seems thus crucial to provide a rigorous evaluation.

- The validation approach describes above only evaluates the anisotropy model "diluted" in a bigger detailed snowpack model. Thus, simulated anisotropy values inherit of all the potential errors from the detailed model. Such evaluation does not allow to identify the causes of deviations, especially what is the contribution of the anisotropy model itself to these deviations.

- No evaluation of the anisotropy model itself is provided. The model consists in three formulations to describe the anisotropy evolution by settlement, by temperature gradient metamorphism, and by wet snow metamorphism. These formulations, although partly based on assumptions, are not assessed. Evaluation is even more crucial for the anisotropy evolution by settlement and melting, processes that have never been clearly shown/supported by quantitative studies, as mentioned by the authors.
What are the relative contributions of these formulations to the anisotropy evolution? How do they compensate each other? How do they "perform", i.e. what is the individual contribution of each formulation to the observed simulation-measurements errors?
Those are some examples of questions to which the paper should definitively be able to answer.

- Thus, before evaluating the model in the frame of a full detailed snowpack model, for long-time period, and for "complex" conditions as encountered in nature, it seems relevant to first assess the model alone, for simple cases of evolution (restricted conditions). To do so, there are few studies on structural anisotropy referred in the paper that would be suitable: controlled experiments where conditions imposed to snow are known and often restricted to few parameters. These experiments could thus be replicated by the anisotropy model, itself, without implement it in a full snowpack model. The works of Schneebeli and Sokratov 2004, Wiese and Schneebeli 2017 or Calonne et al 2014, for example, could be used. I can see that a difficulty might be to deal with the different estimates of anisotropy from the different works, not always based on correlation lengths; solutions could still be find to make relevant comparison. Alternatively, computations based on the set of images of the above mentioned studies (or others) could have been re-do to obtain anisotropy as defined in this paper and allow comparisons.

**2/ Description of the model**
As it is the core of the presented work, the model should be described in more details and evolution laws should be illustrated. More care should be given when presenting previous studies from which the authors partly relied on to built the model.

- I strongly encourage the authors to include a figure that illustrates the anisotropy model by showing how do $A_{strain,}$ $A_{TGM}$ and $A_{melt}$ evolve with time for different values of strain rate, temperature gradient, temperature and liquid water volume fraction (typical min., mean, and max. values for example). This would greatly help apprehending the model: relative contribution of each process, constraints (threshold values) of the model…

- The present model simulate the development of horizontal structures with snow densification. To support such a modelling, the authors provided notably two papers, which seems however a bit over-interpreted or at least would deserve to be described in more details (Section 2.3). Maybe the description of the anisotropy evolution by settlement should appear more clearly like a still hypothetical snow process (since it has never been clearly shown?).

- ○ Schleef and Löwe, 2013
  p.4 l.23: "gravity causes an uniaxial squeeze of the snow structure in the z-direction (Fig. 3 and 4 in Schleef and Löwe, 2013) which increases A". I checked the mentioned figures and, if I am not mistaken, they do not support the above statement (structural anisotropy is not discussed at all in the paper, above mentioned figures highlight the densification process only).
- ○ Wiese and Schneebeli 2017
  Looking at the results of this study (Fig.6), it is actually not really clear how does densification influence anisotropy. For example, why does anisotropy toward horizontal structures develop more in the case of temperature gradient condition with no loading (exp.6) than in the case of isothermal condition with loading (exp.3 and 4)? Why does Exp. 3 and 4 show very little evolution, although the effect of densification should be significant as it is not competing with the opposite effect of temperature gradient?
  It seems that a more detailed descriptions of Wiese and Schneebeli paper would be useful here.

- Regarding the evolution of the anisotropy by densification: what is the role of the initial structure/shape of snow crystals that deposit? Do you expect that plates-shaped crystals (e.g. dendrites) and graupel (I take in purpose two extreme cases) will show the same anisotropy evolution for a given strain rate? The underlying question is: does densification can create horizontal structures in microstructures that were initially isotropic, or only in microstructures that initially present anisotropic crystals shapes. If relevant, it might deserve a comment in the paper.

- A model of the evolution of anisotropy in the case of wet snow is presented. However, the modelling of this specific case is basically only based on assumptions: neither supported by the measurements presented in the paper, which were done only on dry snow, neither by literature studies (no references are given). As a result, there are no evaluations at all of this part of the model, so reader have no clue about the pertinence of the suggested formulation for wet snow metamorphism (eq 14). Thus, the question is, is it relevant at this stage to present wet snow anisotropy at all?

**\*\* Other specific comments (minor)**

- Some ideas are discussed but it is difficult to follow the author's thoughts; they should be reformulated. References are often missing, or it should appear clearly that the authors are talking about hypothesis.

  - ○ p.31 l.4: "Nevertheless, it may surprise that the model completely neglects any dependence on grain size. However, we found that no simple grain size dependence,like weighting the TGM-term by the inverse grain size (by setting the microstructural parameter equal to grain size, $f\,\mu = r\,g$ in Eq. 9), could produce reasonable simulation results. Using the relation $f\,\mu = r\,g$ caused a strong vertical variability of the anisotropy combined with too positive values for the anisotropy of depth hoar (Fig. S24). We still think that neglecting the microstructure could be the main reason why the model was not able to simulate the fast decay of horizontal structures in Jan–Feb 2012." → not clear. Which effect of the grain size do you expect on structural anisotropy? two microstructures with different mean grain sizes but subjected to the same conditions will have different anisotropy rates?

○ p.31 l.11: "Beyond the dimensions of the microstructure, we ignored the crystallographic fabric of snow, i.e. the orientation of the c-axis of the hexagonal ice crystals which compose the microstructure. For the radar data it was ignored because the snow fabric anisotropy affects only very weakly the dielectric anisotropy (Appendix A in Leinss et al. (2016)). For the model, we neither consider the evolution of the snow fabric anisotropy nor the influence of crystal orientation on the evolution on the structural anisotropy. This, because only very few studies exist which provide experimental insight about the orientation of the snow fabric (Calonne et al., 2016) or even the temporal evolution of the snow fabric anisotropy (Riche et al., 2013). Furthermore, the dominant growth direction of snow crystals depends on temperature (Lamb and Hobbs, 1971; Lamb and Scott, 1972) and is not necessarily parallel to the temperature gradient (Miller and Adams, 2009) as it can be clearly observed in the supplementary movie in (Pinzer et al., 2012). Motivated by the competing effect of crystal orientation, structural disorder and structural optimization to increase the vertical thermal conductivity (Staron et al., 2014) we simply introduced a lower limit of the anisotropy $A_{min}$ under TGM." → this paragraph should be reformulated to get clearer. Again, readers need to understand which influence do you expect of the crystalline orientation on structural anisotropy. What is "structural disorder", etc. Beside, this point appears to be a "detail" compared to other assumptions or simplifications of the model. Or do you expect a significant effect?

○ p.16 l.21: "For the initial anisotropy, we neglected any temperature dependence due to lack of representative data. Stronger cohesion between crystals at temperatures close to zero could lead to a more isotropic structure (but with faster settling) compared to cold temperatures were crystals align according to gravity without being influenced by stronger cohesion forces or settling. A temperature dependence for the shape of snow crystals growing in the atmosphere could also influence the initial anisotropy." → here you discuss about potential effect of temperature and crystalline anisotropy, while you do not provided the first basic information that reader would expect, in my view: how "strong" is the assumption of a same initial value for all new layer, i.e. what is the variability of anisotropy of fresh snow (observed/reported)?
+ references are missing to support the described processes

○ p.6 l.27: "Additionally, we assume that horizontal structures in fresh snow decay significantly faster than the growth speed of vertical structures in old snow and add an empirical, quadratic weighting function" → Why? Please explain and/or give references. Besides, the sentence is not clear (do you mean that vertical structures develop faster in fresh snow than old snow? what is old snow?) + incorrect formulation ("structures" cannot decay faster than a "growth speed")

○ p.6 l.10: "The absolute value $|J_v|$ is used because vertical structures can grow independent on the sign of $J_v$" → add references

○ p.6 l.12: "In contrast, temperature gradients changing their direction on a daily scale seem not to increase the anisotropy but cause a rounding of grains (Pinzer and Schneebeli, 2009)." → I could not find any comments on structural anisotropy in Pinzer and Schneebeli 2009. Please provide justifications why oscillating temperature gradient would not cause structural anisotropy (while oscillation longer time period would do).

○ p.7 l.8: "We found, that the model best predicts the measured anisotropy evolution by simply setting $f_\mu(\cdot) = 1$ mm, constant, instead of considering any grain-size dependence. A more physical approach would be to characterize each grain type and size by its potential velocity to transform into vertical structures by a more sophisticated

definition of $f_\mu(\cdot)$. Interestingly, any simple, empirical relation could not produce better results compared to the fixed factor $f_\mu(\cdot) = 1$ mm." → this part is not clear. I do not understand why the authors are interested by modelling the individual growth speed of grains, while willing to describe the structural anisotropy of a layer.

- p.6 l.28: "A faster decay rate of fresh snow compared to old snow partially compensates the fact that any grain size dependence was neglected in the model: the lifetime of small grains in fresh snow should be significantly shorter than the lifetime of large crystals in old snow." → (linked to some above points on influence of grain size) I do not understand this comment.

- p.31 l.4: "It is remarkable how well the model reproduces the radar-measured anisotropy time series." → It is maybe not that remarkable since model is actually calibrated based on the radar measurements to which it is here compared to.

- p.33, l.6: "First the detailed agreement between radar-measured anisotropy and the anisotropy modeled (…) demonstrates that polarimetric radar measurements (…) can be used to monitor the structural evolution of the snow pack". I have a hard time understanding for which application it would be useful to obtain a bulk structural anisotropy (as most snowpack are not homogeneous). The authors should provide concrete ideas of the usefulness.
In the same idea, I am not sure I understood the radar measurements correctly: does a snowpack made of 20 cm of vertical structures (let's say A=-0.2) and 20 cm of horizontal structures (A=0.2) would show a bulk structural isotropy with A=0?

- p.5 l.24: "water molecules diffuse from the bottom up through the ice matrix" → through the pore space?

**\*\* Technical corrections**

- throughout the paper:
  - snow pack → snowpack
  - many errors in the format of citations (brackets or not). Please check.

- p.1, after the 1st sentence, it might be good to recall briefly what is meant by structural anisotropy. "In some cases, snow microstructure can develop a significant structural anisotropy, i.e a structure of ice and air elongated in a particular direction, being often the vertical or horizontal direction." for instance.

- p.2. l.20: add where measurements were done.

- p.3 l.2: "preliminaries" could be replaced by "Defining anisotropy", or similar,  to better reflect the content of the paragraph.

- p.1 l.7: "The model implements..." is not correct. Maybe use "includes"

- p.1 l.23: "and also" → "as well as"

- p.2 l.8: "For snow, the microstructure can be obtained ..."

- p.2 l.14: "The model act as a link" → the formulation seems not correct.

- p.2 l.19: mistake in citation format

- p.3 l.7: mistake in citation format

- p.4 l.1: "...are larger than the vertical scale" → "are larger than the vertical ones".

- p.5 l.11: "for horizontal anisotropies..." → it should be "vertical"?

- p.5 l.16: always

- p.13 l.29: "snow/air images" → "ice/air images"

**References**

Calonne, N., Flin, F., Geindreau, C., Lesaffre, B., and Rolland du Roscoat, S.: Study of a temperature gradient metamorphism of snow from 3-D images: time evolution of microstructures, physical properties and their associated anisotropy, The Cryosphere, 8, 2255–2274, https://doi.org/10.5194/tc-8-2255-2014, http://www.the-cryosphere.net/8/2255/2014/, 2014.

Schneebeli, M. and Sokratov, S.: Tomography of temperature gradient metamorphism of snow and associated changes in heat conductivity, Hydrological Processes, 18, 3655–3665, https://doi.org/10.1002/hyp.5800, http://dx.doi.org/10.1002/hyp.5800, 2004.

Wiese, M. and Schneebeli, M.: Early-stage interaction between settlement and temperature-gradient metamorphism, Journal of Glaciology, 63, 652–662, https://doi.org/10.1017/jog.2017.31, 2017.

---

## Referee Comment (RC2) · Anonymous Referee #2 · 8 Jul 2019

This paper by Leinss et al describes a novel method to obtain anisotropy of snow based on macroscopic properties obtained from snow models, possibly with large impacts on modelling thermal conductivity, structural strength and remote sensing. The paper is very well written and clear to follow, with extensive discussion of the study limitations. The methodology is solid and the authors have usefully connected this work to previous studies that have an alternative definition of anisotropy. This is a clever use of remote sensing combined with snowpack modelling to give insight into the snowpack structure.

Comments and questions:

- Could this model be adapted for an Eulerian snow model (see final question)?

- Pg 8 line 24 add in reference to section 4.1

[Figure]

- Add in introduction (pg 2, line 20) that model is evaluated against micro-CT derived anisotropy.

- Would be useful to put Figure 8 after pg 14, line 10 where it is referenced so it becomes Figure 4.

- Pg 14, line24: What does it mean to enforce the snow height? Pg 30, line 20 states that the snow height is enforced yet is too large.

- Is the 'best' snowpack in Table 3 the one that performs the best over all seasons i.e. same simulation configuration in 2009/2010 as 2010/2011 etc, or the best from each season?

- The purpose of including pc results on pg 21, line 6-13 isn't clear and breaks up the flow of the results. Consider removing them, putting them below line 18 and/or stating here that this is relevant to the discussion.

- Figure 7: could the value of alpha_3 be added to caption to indicate its value relative to alpha_1 and alpha_2?

- It is very hard to distinguish between Tair and Tsoil in Figures 9 and 10. Please change colours and/or line type.

- In the conclusions, pg 33 lines 9 and 21-22 imply that the polarimetric radar measurements are all that are needed to monitor the snow anisotropy. However, a snowpack model will be needed to interpret the anisotropy of the layers so the text should be adjusted accordingly. Extending that a little further, is the role of the CPD then to adjust the relative alpha_1 and alpha_2 per season (cannot be used operationally), is the seasonal fluctuation in these parameters significant (CPD could be used operationally but model needs to be adjusted for Eulerian snowpacks) or are CPD observations needed in the short to medium term to look at different snowpacks / seasons until there is high confidence the snow model can be used to simulate anisotropy without it?

Please correct the following typos:

Pg 5, line 16: alway -> always

Pg 7, line 24: independent on -> independent of

Pg 10, line 16: sensors -> sensor

Pg 13, line 11: sufficiently height -> sufficiently high

Pg 14, line 16: intent -> intend

Pg 15, line 20: settling-induces -> settling-induced

Pg 16, line 5: Finish -> Finnish

Pg 16, line 11: estimate how -> estimate of how

Pg 16, line 23: temperatures were -> temperatures, where

Pg 25, line 18: set it -> set in

Pg 28, line 13: later as -> later than

Pg 32, line 28: folowwing -> following

Pg 33, line 22: non-destructive -> non-destructively

---

## Author Comment (AC1) · 23 Aug 2019

**General comments**

Dear Reviewer #2,

thank you for carefully checking the manuscript, for your constructive comments and for the suggestions to improve the paper. Below are all answers to Reviewer #2. Answers to Reviewer #1 can be found in the other author response.

Prior to the individual answers we like to mention that a model validation suggested by Reviewer #1 could excellently confirm the effect of TGM on the anisotropy evolution. Because of the good agreement we could remove the free parameter $\alpha_2$ from the model and set $\alpha_2 = 1$. As a consequence, only the parameter $\alpha_1$ had to be fitted to the radar

data which allows for a significant simplification and shortening of some sections of the manuscript.

**Answers to Reviewer #2**

**RC 1:** Could this model be adapted for an Eulerian snow model (see final question *[here RC 2 and RC 3]*)?

**AC 1:** In principle yes. As shown in (Krol and Löwe, 2018)), evolution equations for microstructural parameters principally do contain material derivatives which are mainly governed by the settling velocity. By stating a Lagrangian form for the anisotropy evolution, as done here, and superimposing this formulation in SNOWPACK to individual layers, we actually assume implicitly that the Eulerian counterpart of the anisotropy equation is governed by such a Eulerian (PDE) form with a material derivative due to settling. This is the default approach for any microstructural parameter in SNOW-PACK (and any other model utilizing Lagrangian layers) and thus we follow the same approach here. If in contrast a snowpack model would explicitly solve the ice mass continuity equation as an Eulerian conservation law with an advective settling term, the evolution equations for the anisotropy should follow the same procedure.
As the model describes the temporal evolution of individual snow layers we do not see much advantage in formulating it in a Eularian coordinate system. To support this, we like to refer to the four remarks on p. 127 in (Bartelt and Lehning, 2002), especially point 4) which states: "It is impossible to track snow microstructure parameters using a Eulerian formulation since material history is lost.". As we model explicitly the material history an Eulerian formulation would not be compatible with our model.
To point this out more clearly, we suggest to add the above reference to Section 2.2 where we address the Lagrangian viewpoint: "As common for snow models focusing on the evolution of properties of individual snow layers (Bartelt and Lehning, 2002), we describe(...)"

*Please note that we moved the final question of reviewer #2 here (RC2 and RC3).*

**RC 2:** In the conclusions, p.33 lines 9 and 21-22 imply that the polarimetric radar measurements are all that are needed to monitor the snow anisotropy. However, a snowpack model will be needed to interpret the anisotropy of the layers so the text should be adjusted accordingly.

**AC 2:** Polarimetric radar measurements are all that is needed to *observe* the *depth-averaged* snow anisotropy. However, a snowpack model is needed to *model* the anisotropy of *individual layers* which cannot be measured with polarimetric radar systems, at least not without tomographic radar imaging methods. (See also AC20 for Reviewer #1)
We suggest to make clear in line 9 that with radar remote sensing systems only the *depth-averaged* anisotropy can be measured but we like to point out that with in-situ systems even the depth-resolved evolution of the anisotropy could be measured. In the current words "(...) to monitor the structural evolution of the snow pack", "structural" might be misunderstood as layer-wise. To clarify we suggest to write: "(...) to monitor the evolution of the structural anisotropy the snow pack." And add "Depending on the system geometry the anisotropy can be measured only depth-averaged (remote-sensing systems) or even depth-resolved with in-situ systems (Fujita et al., 2009)." With these text adjustments also line 21-22 should read clearer.

**RC 3:** Extending that [RC2] a little further, is the role of the CPD then to adjust the relative $\alpha_1$ and $\alpha_2$ per season (cannot be used operationally), is the seasonal fluctuation in these parameters significant (CPD could be used operationally but model needs to be adjusted for Eulerian snowpacks) or are CPD observations needed in the short to medium term to look at different snowpacks / seasons until there is high confidence the snow model can be used to simulate anisotropy without it?

**AC 3:** As mentioned in the general comment, the validation of the TGM equation of

the model with CT data confirmed that $\alpha_2 = 1$ is very likely valid for all kinds of snow. However, due to the lack of data sets including anisotropy time series of fresh snow under settling we need to restrict the validity of the parameters $\alpha_1$ to the test site in Sodankylae and the years 2009-2013. However, the similarity of obtained values of $\alpha_1$ for individual seasons (Figure 7 and Table 5, TCD) indicates that the same values can be used for all seasons. All our main results (Fig. 4, 5, 6, TCD) are based on constant values for $\alpha_1$ and $\alpha_2$ (in the revision, this will only be $\alpha_1$) which value(s) have been determined from the complete set of data spanning all four seasons.

To clarify this, we suggest to adjust p.33, line 2 (TCD) to "free parameters (...) calibrated by globally minimizing the difference between (...) and four years of anisotropy data (...). The radar data were acquired in Sodankylä, Finland between 2009 and 2013." In addition, as requested by Reviewer #1, we will include an explicit validation of the temperature gradient term in the evolution equation that confirms (independent of CPD measurements) the correctness at least of one part of the model.

**RC 4:** Pg 8 line 24 add in reference to section 4.1.

**AC 4:** Fine. We suggest to adjust the text to: "SNOWPACK was calibrated (...) by snow temperature (Sect. 4.1)."

**RC 5:** Add in introduction (p.2, line 20) that model is evaluated against micro-CT derived anisotropy.

**AC 5:** Sure. Additionally we will add the validation against independent CT data. We suggest to add: "The TGM formulation of the model is validated with independent CT laboratory experiments. The full model is evaluated against full-depth micro-CT derived anisotropy profiles from the field".

**RC 6:** Would be useful to put Fig 8 after p.14, line 10 where it is referenced so it becomes Figure 4.

**AC 6:** In principle, I would agree. However, Section 5.1 (Results: CT validation) and especially Section 6.1 (Discussion of model resuls) refer quite frequently to Figure 8. Therefore, I prefer to keep it close to Section 6.1 and accept the not-in-order reference to Figure 8.

**RC 7:** P.14, line 24: What does it mean to enforce the snow height? P.30, line 20 states that the snow height is enforced yet is too large.

**AC 7:** we suggest to adjust the text to: "(...) enforced snow height, i.e. SNOWPACK models the amount of precipitation based on measured snow height". To make clear, that SNOWPACK failed to enforce snow height in Nov 2012, we suggest to adjusted P.30, line 20: "Because snow height was enforced, the too large snow height end of Nov implied forcing with too low precipitation for mid of Dec which resulted in less fresh snow with a positive anisotropy and in turn explains why the simulated anisotropy is lower than the radar-measured anisotropy, Fig. 5(d)." to "Because SNOWPACK struggled enforcing a decreasing snow height in late Nov, the subsequent amount of fresh snow required to match the enforced snow height early Dec was underestimated. The missing effect of settling of fresh snow explains why the simulated anisotropy in Dec 2012 is lower than the radar-measured anisotropy, Fig.5(d)."

**RC 8:** Is the 'best' snowpack in Table 3 the one that performs the best over all seasons i.e. same simulation configuration in 2009/2010 as 2010/2011 etc, or the best from each season?

**AC 8:** It's the best over all four seasons. To make this clearer, we suggest to adjust p. 14 line 20 to "(...) we run for all four seasons more than 5000 simulations with each time different settings (but keeping the same settings for all four seasons)..." and also p. 14, line 25 to "Details (...) and the definition (grading) of the 'best' set of simulations (with same settings for all four seasons) are described in Appendix A3.

**RC 9:** The purpose of including $p_c$ results on p.21, line 6-13 isn't clear and breaks up the flow of the results. Consider removing them, putting them below line 18 and/or stating here that this is relevant to the discussion.

**AC 9:** As the discussion (6.4) goes quite into detail of uncertainties of microstructure characterization we think that it's worth to mention the comparison to $p_c$. However, we moved (and slightly shortened) this paragraph below line 18 and added a reference to the discussion.

**RC 10:** Figure 7: could the value of $\alpha_3$ be added to caption to indicate its value relative to $\alpha_1$ and $\alpha_2$?

**AC 10:** We suggest to add to the caption "Other parameters ($A_{min} = -0.6$; $A_{max} = 0.3$, $A_{ini}$ = 0.05 and $\alpha_3 = 3 \cdot 10^{-5}$) were kept constant." Please note that $A_{min}$ has been changed from $-0.3$ to $-0.6$ to provide a better agreement with the independent CT validation data.

**RC 11:** It is very hard to distinguish between $T_{air}$ and $T_{soil}$ in Figures 9 and 10. Please change colours and/or line type.

**AC 11:** We suggest to change the style of $T_{soil}$ to dotted lines, the colors to black/gray and will draw them ontop of $T_{air}$ which makes $T_{air}$ and $T_{soil}$ much better distinguishable.

**technical corrections:**

**RC 12:** Please correct the following typos:
**AC 12:** The 13 typos listed by Reviewer #2 will be corrected.

**References**

Bartelt, P. and Lehning, M.: A physical SNOWPACK model for the Swiss avalanche warning: Part I: numerical model, Cold Regions Science and Technology, 35, 123 – 145, https://doi.org/http://dx.doi.org/10.1016/S0165-232X(02)00074-5, 2002.

Fujita, S., Okuyama, J., Hori, A., and Hondoh, T.: Metamorphism of stratified firn at Dome Fuji, Antarctica: A mechanism for local insolation modulation of gas transport conditions during bubble close off, Journal of Geophysical Research: Earth Surface, 114, 1–21, https://doi.org/10.1029/2008JF001143, 2009.

Krol, Q. and Löwe, H.: Upscaling ice crystal growth dynamics in snow: Rigorous modeling and comparison to 4D X-ray tomography data, Acta Materialia, 151, 478 – 487, https://doi.org/https://doi.org/10.1016/j.actamat.2018.03.010, 2018.

---

## Author Comment (AC2) · 23 Aug 2019

**General comments**

Dear Reviewer #1,

thank you for your constructive comments and suggestions for improving the paper. Below we will answer all comments by Referee #1. Answers to Referee #2 can be found in the other author response.

In summary, we will include an explicit validation of the evolution of anisotropy under temperature gradients from CT experiments that allows us to eliminate one of the free parameters. This requires a slight extension of the paper in this direction but at the same time we can shorten the paper at many other places. Detailed answers are given

below.

**Answers to Reviewers #1:**

**RC 1:** (...) It [the paper] is well written, although it would gain clarity by being shortened or more to the point in some paragraphs.

**AC 1:** We will check the paper carefully and shorten it where ever possible and will concisely rewrite some paragraphs. As we suggest below, the validation of the TGM term of the model makes the fit-parameter $\alpha_2$ superfluous and in consequence a lot of text about determination of the two parameters $\alpha_1$ and $\alpha_2$ can be removed which should significantly shorten the paper. Additionally to that, we try to shorten the following paragraphs:
- Results based on pc (pg 21, line 3-12) will be reformulated/moved. See also answer to RC9 by Reviewer 2.
- The discussion section will be shortened.
- The appendix will be shortened.

**Specific comments** (major).

*The following major comments RC 2 and RC 3 are split in several subcomments which are answered separately.*

**RC 2:** Robust evaluation is missing:
The validation step of the anisotropy model is too little addressed in the paper. The evolution of structural anisotropy has been only observed in some cases, and is not fully understood yet, especially the evolution during settlement and wet snow metamorphism. The proposed model relies thus partly on hypothetical processes. It seems thus crucial to provide a rigorous evaluation.
**AC 2:** Generally we agree with Reviewer1, that the individual contributions of the model (TGM, settling, melt metamorphism) should be validated before running the full model on long time series where effects of individual contributions cannot be assessed. However, as validation data is rare we can only validate the TGM equation and follow the Reviewer's suggestion to distinguish more clearly what is based on assumptions and on observations.

For example, we suggest to clearly identify the equations for snow settling and melt metamorphism as a *hypothesis*.

**AC 2.1:** To address the missing robust evaluation we will comment separately on the individual contributions below, keeping in mind that the additive superposition is already a simplification and suggest to cite (Wiese and Schneebeli, 2017b) on p.4, line 12 (TCD) who showed that the strain rate is coupled to temperature gradients (We suggest also to comment on this observations when stating that "Naturally, in snow all these processes are coupled (section 2.1)").

**AC 2.2:** Validation of anisotropy evolution during TGM:
For temperature gradient metamorphism (TGM), we will include an explicit validation using different sets of existing laboratory data (TGM-2, TGM-17, DH-1, DH-2) analyzed by (Löwe et al., 2013) and additional data labeled C-1 from (Calonne et al., 2014), which were acquired for snow at different ice volume fractions (0.22, 0.33, 0.19, 0.29, 0.35), temperatures (-10, -8, -20, -20, -4C) and temperature gradients (100, 50, 50, 50, 43 K/m). For all these data the growth of vertical structures during TGM can be compared with the model. Fig 1(a) shows the evolution of the observed anisotropy, dashed lines indicate modeled results. Interestingly, in an early stage a few days after sample preparation, Fig 1.(b) shows that the anisotropy seems to be quite stable before vertical structures grow. When ignoring the limiting factor $(A - A_{min})^2/A^2_{min}$ in Eq. (9, TCD) and setting $\alpha_2 = 1$ one obtains the anisotropy evolution by time integration $A_{TGM}(t) = A(0) + |J_v|/(\rho_{ice} * f_\mu)t$, which already agrees well with the experimental data as shown in Fig. 1(c). This figure indicates that the growth of vertical structures

is proportional to the water vapor flux $J_v$ as modeled by Eq. (13, TCD) for different temperatures and temperature gradients (will be listed in a Table, here Fig. 2). Adding the limiting factor $(A - A_{min})^2/A_{min}^2$ to the model and using setting $A_{min} = -0.6$ further improves the correlation with the laboratory data, Fig. 1(d).

Because of the good agreement shown in Fig. 1(d) the "free" parameter $\alpha_2$ (for TGM) can be eliminated and set constant to 1.0 as confirmed by the CT time series. As in this case we do not require any extended discussion of the parameter set $\alpha_1$ and $\alpha_2$ and because $\alpha_1$ (for snow settling) is then the only fit parameter, the paper can be simplified and shortened at many places.

**AC 2.3:** Validation of anisotropy evolution by snow settling:
For the evolution of anisotropy under settling presently no conclusive validation can be given without extensive re-evaluation of existing data or conducting new, tailored experiments: As stated in the TCD paper (Sec 6.4) the evaluation of the anisotropy for the dielectric tensor in the present form (based on the correlation lengths) relies on the assumption of spheroidal symmetry of the correlation function. Strictly this assumption is apparently violated in new snow under settling as a direct consequence of the observed differences in the evolution of different length scales in the correlation function (Löwe et al., 2011). This implies that presently we can only effectively cast the settling contribution into the given form, when using the exponential correlation lengths to evaluate the anisotropy. This will be stated more explicitly.

For the settling induced "growth" of horizontal structures we will point out clearly, that the evidence from lab-experiments is ambivalent anyway:. e.g. (Wiese and Schneebeli, 2017b) did not observe an increase of initially existing horizontal structures in sintered snow of relatively high density (200...300 kg m$^{-3}$) during isothermal settlement, but others observed either a squeeze in fresh snow of low density (100 kg m$^{-3}$) (Schleef and Löwe, 2013) or an increase of horizontal structures a few days after fresh snow deposition (Leinss et al., 2016). We will point out this more clearly and suggest to rewrite the first paragraph of section 2.3 (gravitational settling), p. 4, line 20 - 24 as

suggested below:

The first term in Eq.(3), $A_{strain}$(t), accounts for gravitational settling and densification of snow which have *assumed* to be the cause of horizontal structures in polarimetric radar data. Interestingly, in these data the horizontal structures did not appear instantaneously with fresh snow but with a time delay of a few days after snow fall, thereby suggesting a settling effect (Leinss et al., 2016, Sect. 5.4). However, such growth of horizontal structures during compaction could not be observed in snow which has sintered for several months after initial sample preparation by sieving (Wiese and Schneebeli, 2017b). Nevertheless, in this work most samples showed a slight horizontal structure at the beginning of the experiment which, in combination with the observation in (Leinss et al., 2016), rises the *hypothesis* that a horizontal anisotropy grows only in an initial phase after snow deposition, likely due to settling. In contrast to (Wiese and Schneebeli, 2017b), (Schleef and Löwe, 2013) avoided any sintering with the aim to study new snow and confirmed "the anisotropic nature of densification" by attributing density changes "solely to a squeeze of the structure in the vertical direction, i.e. to axial strains". Such a squeeze can be observed in (Schleef and Löwe, 2013, Fig. 4) where the "displacement of characteristic maxima and minima of the density profile" at different times is indicated by arrows. The relative position of the arrows to each other shows that "the structure is strained during 2 days of densification". From that we draw the *hypothesis* that the ice matrix is squeezed, at least in an initial state of fresh snow.

Additionally, because of a lack of CT data showing the anisotropy evolution of *fresh* snow during settling (the snow in (Wiese and Schneebeli, 2017b) has sintered for several months) we suggest to point out in the discussion that for a proper evaluation of this model part further experiments would be required which is far beyond the scope of this study.

**AC 2.4:** validation of wet snow metamorphism
For wet snow metamorphism the situation for independent experimental validation is even worse and we suggest to provide a comparison in the supplementary figures of

the model when the wet snow metamorphism term switched off. (Figures 3 and 4). Additionally we like to add a short explanation to the paper "that the surface tension of water should cause a rounding of ice grains by melt metamorphism and that an anisotropic structure would be driven therefore towards isotropy." Similar to (Lehning et al., 2002) and (Brun et al., 1992), who both use similar equations, we will clearly stress that this part is presently based on an unconfirmed assumption.

**RC 2.5:** Thus, before evaluating the model in the frame of a full detailed snowpack model, for longtime period, and for "complex" conditions as encountered in nature, it seems relevant to first assess the model alone, for simple cases of evolution (restricted conditions).

**AC 2.5:** See evaluation and comments (AC 2.1 - AC 2.4) above. We hope that comparably high standards will be applied for any parametrization in snowpack models in the future.

**RC 2.6:** To do so, there are few studies on structural anisotropy referred in the paper that would be suitable: controlled experiments where conditions imposed to snow are known and often restricted to few parameters. These experiments could thus be replicated by the anisotropy model, itself, without implement it in a full snowpack model. The works of (Schneebeli and Sokratov, 2004; Wiese and Schneebeli, 2017b) or (Calonne et al., 2014), for example, could be used. I can see that a difficulty might be to deal with the different estimates of anisotropy from the different works, not always based on correlation lengths; solutions could still be find to make relevant comparison.

**AC 2.6:** As describe above, we used data analyzed by (Löwe et al., 2013) acquired by (Kaempfer et al., 2005; Riche et al., 2013; Löwe et al., 2013) and data from (Calonne et al., 2014) as suggested by the Reviewer. The data of (Schneebeli and Sokratov, 2004) was not used because of the difference in anisotropy metric (it provides the "degree of anisotropy" (without sign)) and the experiments are similar to the other listed

studies. The work of (Wiese and Schneebeli, 2017b) could not be used for a quantitative analysis because of i) the difference in the anisotropy metric and ii) the used sintered snow of significantly higher density which is not comparable to fresh snow. However, the work of (Wiese and Schneebeli, 2017b) shows an important limitation of the anisotropy model as well as of SNOWPACK which both do not consider the strong coupling of TGM and the settling rate as observed by (Wiese and Schneebeli, 2017a). To account for this, we suggest to comment on this in the discussion:
SNOWPACK does not consider the coupling of TGM and the settling rate as observed by (Wiese and Schneebeli, 2017a). A modification of the settling rate in SNOWPACK would affect the anisotropy evolution of fresh snow.

**RC 2.7:** Alternatively, computations based on the set of images of the above mentioned studies (or others) could have been re-do to obtain anisotropy as defined in this paper and allow comparisons.

**AC 2.7:** We like to suggest this idea in the discussion by addressing settling of snow and will cite the work of (Schleef and Löwe, 2013) and (Schleef et al., 2014). However, re-analyzing this entire dataset would be to time- and cost intense and therefore definitely beyond the scope of this paper.

**RC 3:** Description of the model
As it is the core of the presented work, the model should be described in more details and evolution laws should be illustrated. More care should be given when presenting previous studies from which the authors partly relied on to built the model.

**AC 3:** Several references have been checked as suggested by Reviewer 1 (see comments below).

**RC 3.1:** I strongly encourage the authors to include a figure that illustrates the anisotropy model by showing how do Astrain, ATGM and Amelt evolve with time for

different values of strain rate, temperature gradient, temperature and liquid water volume fraction (typical min., mean, and max. values for example). This would greatly help apprehending the model: relative contribution of each process, constraints (threshold values) of the model...

**AC 3.1:** Though even lengthening the paper, we suggest to add Fig. 5 and the corresponding table to the result section. Fig. 5 would also help to reproduce our model results and could act as a verification when re-implementing the model. In the figure, each line is labeled with a number; the corresponding parameters are listed in the table below. We will add some discussion to the figure which illustrates on which time scales different processes happen: e.g. TGM runs on completely different time scales (longer than 10 days) compared to settling and melt metamorphism (faster than 10 days). Additionally, Fig. 5(a) shows that the water vapor flux varies by about 1 order of magnitude for snow temperatures between -20 and 0C and depends linearly on the temperature gradient.

**RC 3.2:** The present model simulate the development of horizontal structures with snow densification. To support such a modeling, the authors provided notably two papers, which seems however a bit over-interpreted or at least would deserve to be described in more details (Section 2.3). Maybe the description of the anisotropy evolution by settlement should appear more clearly like a still hypothetical snow process (since it has never been clearly shown?).

**AC 3.2:** Yes, we totally agree and will clearly identify it as a *hypothesis*. See also AC 2.3 and AC 3.

**RC 3.3:** (Schleef and Löwe, 2013) p.4 l.23: "gravity causes an uniaxial squeeze of the snow structure in the z-direction (Fig. 3 and 4 in (Schleef and Löwe, 2013)) which increases A". I checked the mentioned figures and, if I am not mistaken, they do not support the above statement (structural anisotropy is not discussed at all in the paper,

above mentioned figures highlight the densification process only).

**AC 3.3:** The paper of (Schleef and Löwe, 2013) supports the statement of an "uniaxial squeeze" which should increase the anisotropy according to our model. We explained our conclusion from this paper already in detail in AC 2.2.

**RC 3.4:** (Wiese and Schneebeli, 2017b) Looking at the results of this study (Fig. 6), it is actually not really clear how does densification influence anisotropy. For example, why does anisotropy toward horizontal structures develop more in the case of temperature gradient condition with no loading (exp.6) than in the case of isothermal condition with loading (exp.3 and 4)? Why does Exp. 3 and 4 show very little evolution, although the effect of densification should be significant as it is not competing with the opposite effect of temperature gradient? It seems that a more detailed descriptions of (Wiese and Schneebeli, 2017b) paper would be useful here.

**AC 3.5:** Indeed, Figure 6 in (Wiese and Schneebeli, 2017b) does not support our hypothesis that densification influences the anisotropy. However, as discussed in AC 2.3 (Wiese and Schneebeli, 2017b) studied sintered snow of a relative high density which cannot be compared to fresh snow. We therefore suggest to restrict our assumption "that settling increases the anisotropy" to fresh snow which is supported by (Schleef and Löwe, 2013), and the observations in (Leinss et al., 2014) and (Leinss et al., 2016).

**RC 3.6:** Regarding the evolution of the anisotropy by densification: what is the role of the initial structure/shape of snow crystals that deposit? Do you expect that plates-shaped crystals (e.g. dendrites) and graupel (I take in purpose two extreme cases) will show the same anisotropy evolution for a given strain rate? The underlying question is: does densification can create horizontal structures in microstructures that were initially isotropic, or only in microstructures that initially present anisotropic crystals shapes. If relevant, it might deserve a comment in the paper.

**AC 3.6:** As mentioned in the last paragraph of AC 2.3 and AC 2.7, no CT data about

densification of fresh snow and the corresponding anisotropy is available to us. This includes of course any microscopic information about snow type or crystal shape and the related behavior of the anisotropy under densification. Therefore it is beyond the scope of this study to answer this question. However, we will come back to this point later (AC 4.3 and AC 4.7a) where we provide arguments that the initial anisotropy should not very too much (less than a range of about $A\pm0.05$) and that we ignored any microstructure by setting the microstructural parameter in our model fixed to $f_\mu = [1]mm$.

**RC 3.7:** A model of the evolution of anisotropy in the case of wet snow is presented. However, the modeling of this specific case is basically only based on assumptions: neither supported by the measurements presented in the paper, which were done only on dry snow, neither by literature studies (no references are given). As a result, there are no evaluations at all of this part of the model, so reader have no clue about the pertinence of the suggested formulation for wet snow metamorphism (eq 14). Thus, the question is, is it relevant at this stage to present wet snow anisotropy at all?

**AC 3.7:** I think we should comment about the expected behavior of the anisotropy of wet snow under melt metamorphism as suggested in AC 2.4 but will clearly state that this part of the model is completely based on reasonable assumptions. In our dataset, the influence of melt metamorphism affects almost only the snow melt period for which no radar data is available. Still, the short melt event in April 2012 after which the snow pack refroze provides a few data points which support the *qualitative* observation that melt metamorphism decreases the anisotropy. We will therefore make clear in the discussion that further studies are needed to provide *quantitative* data on melt metamorphism. To show that melt metamorphism does not affect our model we like to provide a comparison of our model with and without the melt metamorphism part (Fig. 3 vs. Fig. 4) and suggest to add the figure with melt metamorphism switched off to the supplementary material. Note that in both figures we set $\alpha_2 = 1.0$, $A_{min} = -0.6$ and determined $\alpha_1$ by fitting the model to the radar data.

**RC 4:** Other specific comments (minor)
Some ideas are discussed but it is difficult to follow the author's thoughts; they should be reformulated. References are often missing, or it should appear clearly that the authors are talking about hypothesis.

**AC 4:** We will answer below the specific comments to address this general comment.

**RC 4.1:** p.31 l.4: "Nevertheless, it may surprise that the model completely neglects any dependence on grain size. (...) We still think that neglecting the microstructure could be the main reason why the model was not able to simulate the fast decay of horizontal structures in Jan-Feb 2012."
not clear - Which effect of the grain size do you expect on structural anisotropy? Two microstructures with different mean grain sizes but subjected to the same conditions will have different anisotropy rates?

**AC 4.1:** Yes, we would expect that under the same conditions, e.g. the grain size should affect the anisotropy change rate. However, we could not find any improvement of the model results when implementing a grain size dependence. Notably, when using $A_{min}$ = -0.6 instead of $A_{min} = -0.3$ as used in the TCD manuscript, the model agrees well with the CT validation data and is able to reproduce the fast decay of the horizontal structures in Jan-Feb 2012 which was not possible in the TCD manuscript. We will adjust the relevant section accordingly. (see next comment)

**RC 4.2:** p.31 l.11: "Beyond the dimensions of the microstructure, we ignored the crystallographic fabric of snow, i.e. the orientation of the c-axis of the hexagonal ice crystals which compose the microstructure. For the radar data it was ignored because the snow fabric anisotropy affects only very weakly the dielectric anisotropy (Appendix A in (Leinss et al., 2016)). For the model, we neither consider the evolution of the snow fabric anisotropy nor the influence of crystal orientation on the evolution on the structural anisotropy. This, because only very few studies exist which provide experimental insight about the orientation of the snow fabric (Calonne et al., 2016) or even
the temporal evolution of the snow fabric anisotropy (Riche et al., 2013). Furthermore,
the dominant growth direction of snow crystals depends on temperature (Lamb and
Hobbs, 1971; Lamb and Scott, 1972) and is not necessarily parallel to the temperature
gradient (Miller and Adams, 2009) as it can be clearly observed in the supplementary
movie in (Pinzer et al., 2012). Motivated by the competing effect of crystal orientation,
structural disorder and structural optimization to increase the vertical thermal conduc-
tivity (Staron et al., 2014) we simply introduced a lower limit of the anisotropy $A_{min}$
under TGM."
this paragraph should be reformulated to get clearer.
- Again, readers need to understand which influence do you expect of the crystalline
orientation on structural anisotropy. What is "structural disorder", etc.
- Beside, this point appears to be a "detail" compared to other assumptions or simplifi-
cations of the model. Or do you expect a significant effect?

**AC 4.2:** This point is indeed a detail but still important to mention. Therefore we
suggest to reformulate it in such a way that it is clear to the reader why we ignore the
crystallographic fabric.

**RC 4.3:** p.16 l.21: "For the initial anisotropy, we neglected any temperature depen-
dence due to lack of representative data. Stronger cohesion between crystals at tem-
peratures close to zero could lead to a more isotropic structure (but with faster settling)
compared to cold temperatures were crystals align according to gravity without be-
ing influenced by stronger cohesion forces or settling. A temperature dependence for
the shape of snow crystals growing in the atmosphere could also influence the initial
anisotropy."
here you discuss about potential effect of temperature and crystalline anisotropy, while
you do not provided the first basic information that reader would expect, in my view:
how "strong" is the assumption of a same initial value for all new layer, i.e. what is the
variability of anisotropy of fresh snow (observed/reported)? + references are missing

to support the described processes.

**AC 4.3:** We suggest to provide the basic information first that "we expect that the initial anisotropy after snow fall should not vary much around zero but assume that it should be slightly positive because of some initial settling. Due to lack of representative data we ignore any temperature dependence..." I think this paragraph can also be shortened. For the temperature dependent shape of crystals we suggest to cite (Libbrecht, 2005).

**RC 4.4:** p.6 l.27: "Additionally, we assume that horizontal structures in fresh snow decay significantly faster than the growth speed of vertical structures in old snow and add an empirical, quadratic weighting function"
Why? Please explain and/or give references. Besides, the sentence is not clear (do you mean that vertical structures develop faster in fresh snow than old snow? what is old snow?) + incorrect formulation ("structures" cannot decay faster than a "growth speed").

**AC 4.4:** We suggest to reformulate this sentence to: "Because small ice grains in fresh snow evaporate faster than large ice grains in older snow, we assume additionally that horizontal structures decay significantly faster than vertical structures can grow and add therefore an empirical, quadratic weighting function."

**RC 4.5:** p.6 l.10: "The absolute value $|J_v|$ is used because vertical structures can grow independent on the sign of $J_v$
add references

**AC 4.5:** We suggest to make clear that this is an assumption based on the argument that "Snow crystals always grow in the opposite direction of the vapor flux Pinzer et al. (2012); Yosida (1955). Because the anisotropy does not contain information about the growth direction but only on the growth orientation, we assume that the growth of

vertical structures can be modeled proportional to the absolute value $|J_v|$." (See also next answer for RC 4.6)

**RC 4.6:** p.6 l.12: "In contrast, temperature gradients changing their direction on a daily scale seem not to increase the anisotropy but cause a rounding of grains (Pinzer and Schneebeli, 2009)."
I could not find any comments on structural anisotropy in (Pinzer and Schneebeli, 2009). Please provide justifications why oscillating temperature gradient would not cause structural anisotropy (while oscillation longer time period would do).

**AC 4.6:** Our current formulations may read a bit like an over-interpretation of the results in (Pinzer and Schneebeli, 2009). Therefore we suggest to reformulate this sentence as: "In contrast, temperature gradients changing their direction with a daily cycle seem not to cause the growth of faceted crystals: according to (Pinzer and Schneebeli, 2009) the morphology of the snow structure evolves much slower under alternating temperature gradients and did not show any sign of TGM. Therefore we exclude the effect of daily alternating temperature gradients."

**RC 4.7a:** p.7 l.8: "We found, that the model best predicts the measured anisotropy evolution by simply setting $f_\mu(\cdot)$ = 1 mm, constant, instead of considering any grain-size dependence. A more physical approach would be to characterize each grain type and size by its potential velocity to transform into vertical structures by a more sophisticated definition of $f_\mu(\cdot)$. Interestingly, any simple, empirical relation could not produce better results compared to the fixed factor $f_\mu(\cdot)$ = 1 mm."
this part is not clear. I do not understand why the authors are interested by modelling the individual growth speed of grains, while willing to describe the structural anisotropy of a layer.

**AC 4.7a:** Without any limitation, we would expect that the time a structure needs to transform into a more vertical structure should depend on its microstructure, which
can be described e.g. by grain size, SSA, or other structural parameters. Therefore, the anisotropy change rate should be somehow related to the grain size growth rate because both depend on the transport of water molecules. We will try to point this out more clearly.

**RC 4.7b:** p.6 l.28: "A faster decay rate of fresh snow compared to old snow partially compensates the fact that any grain size dependence was neglected in the model: the lifetime of small grains in fresh snow should be significantly shorter than the lifetime of large crystals in old snow."
(linked to some above points on influence of grain size) I do not understand this comment.

**AC 4.7b:** We hope, that AC 4.7a already clarified that a potential connection between the anisotropy change rate and the grain size growth rate should exist. We suggest therefore to rephrase "A faster decay rate of fresh snow" to "A faster decay rate of the anisotropy of fresh snow".

**RC 4.8:** p.31 l.4: "It is remarkable how well the model reproduces the radar-measured anisotropy time series."
It is maybe not that remarkable since model is actually calibrated based on the radar measurements to which it is here compared to.

**AC 4.8:** The agreement between model and radar data needs to be seen in the context that four years of radar time series, consisting of over 3000 different data points, could be reproduces by fitting the limited set of around 2-5 parameters. With the additionally provided validation using independent CT data (see AC 2.2 and the corresponding figure 1) we could even reduce the number of free parameter by one. At the same time our results still agree with the radar data and even better with the in-situ CT validation data. We think that this is very remarkable.

**RC 4.9:** p.33, l.6: "First the detailed agreement between radar-measured anisotropy and the anisotropy modeled (...) demonstrates that polarimetric radar measurements (...) can be used to monitor the structural evolution of the snow pack".

I have a hard time understanding for which application it would be useful to obtain a bulk structural anisotropy (as most snowpack are not homogeneous). The authors should provide concrete ideas of the usefulness.

In the same idea, I am not sure I understood the radar measurements correctly: does a snowpack made of 20 cm of vertical structures (let's say A=-0.2) and 20 cm of horizontal structures (A=0.2) would show a bulk structural isotropy with A=0?

**AC 4.9:** With standard polarimetric radar systems only the bulk anisotropy of the snow pack can be observed, such that your example above would indeed result in A=0 (see also answer to RC 2 by Reviewer #2). However, as shown in (Leinss et al., 2014) by observing time series of the anisotropy some information about the amount and the timing of snow fall events can be extracted. With in-situ radar systems which have antennas moving on a rail, the snowpack could be scanned layer by layer, either by using methods of radar tomography or with antennas transmitting horizontally through different layers of the snow pack. We will provide such examples. (See also answer AC 2 to Reviewer #2).

**RC 4.10:** p.5 l.24: "water molecules diffuse from the bottom up through the ice matrix" through the pore space?

**AC 4.10:** Yes, we suggest to write that they will "diffuse through the pore space surrounding the ice matrix".

**RC 5:** Technical corrections

**AC 5:** All technical corrections will be applied.

**References**

[revised manuscript text omitted]

the Institute of Low Temperature Science, 7, 19, 1955.

[Figure]

[Figure]

**Fig. 1.** (a) anisotropy time series from lab experiments; (b) first 15 days of anisotropy evolution; (c): CT data vs. time integration of Jv; (d): CT data vs. full model.

| sample | T °C | $\nabla T$ K/m | $f_{\mathrm{v}}(0)$ - | type - | SSA $\mathrm{m^2\ kg^{-1}}$ | $\Delta t$ days |
|--------|------|-----|---------|------|-----|-----|
| TGM-2  | -10  | 100 | 0.22    | DFdc | 29.0 | 11.7 |
| TGM-17 | -8   | 50  | 0.33    | RGsr | 21.7 | 16.0 |
| DH-1   | -20  | 50  | 0.19    | DFdc | 22.1 | 87.5 |
| DH-2   | -20  | 50  | 0.29    | DFbk | 20.0 | 80.5 |
| C-1    | -4   | 43  | 0.35    | RG   | 20.8 | 27.7 |

**Fig. 2.** Experimental conditions for TGM experiments.

[Figure]

**Fig. 3.** Results of the full model with $\alpha_1 = 1.0$, $\alpha_2 = 2.8$, $\alpha_3 = 3e\text{-}6$, Amin=-0.6, Amax=0.3

[Figure]

**Fig. 4.** Same as Fig. 3 but without melt metamorphism.

[Figure]

**Fig. 5.** The evaluation of different parts of the model show that TGM and settling evolve on different time scales.

---

## Author Response (AR1)

Dear **Reviewer #1** and **Reviewer #2,**

We thank you for your constructive comments and suggestions for improving the paper. We considered all comments carefully and reworked the manuscript substantially. Especially the suggestion by Reviewer #1 to validate the paper more carefully with additional datasets lead to major changes of the manuscript. This did not only further improve and confirm the model but it also helped us to shorten the paper significantly, by in total four pages in the final two-column format.

Below we like to list the major changes of the manuscript:

– Five additional datasets from independent CT experiments were added for validation of the model formulation for temperature gradient metamorphism (TGM).

– These added CT data validated and confirmed the TGM formulation of the model and allowed to determined the two free parameters $\alpha_1$ and $A_{min}$ independently from the radar measured anisotropy time series.

– Because $\alpha_1$ (for TGM) was determined independently, the other free parameter $\alpha_2$ (to describe the settling-induced growth of horizontal structures) could be determined from radar data after $\alpha_1$ has already been fixed by CT data.

– The reduction of the solution space of the free parameters from two dimensions $(\alpha_1, \alpha_2)$ to a single dimension $(\alpha_2)$ and a slight correction of the radiation data allowed for determination of the free parameter $\alpha_2$ simply by minimizing the root-mean-square error between model and radar anisotropy time series. This allowed us to remove the entire section about the cost function to determine the free parameters.

– Because the two parameters $\alpha_1$ and $\alpha_2$ are now determined independently, several paragraphs which previously discussed their relation could be removed.

– To avoid confusion we like to note here that the definitions of $\alpha_1$ and $\alpha_2$ have been swapped compared to the initially submitted TCD manuscript.

– Additionally, we combined the two sections *Data sets and testsite* and *Methods: model forcing and calibration* into a single section *Datasets and methods* which improved the flow of the paper and helped to shorten many paragraph.

– The model part about melt metamorphism was removed and is now only suggested as a speculative part in the discussion.

– A section about the model evaluation has been added.

– Two full page figure which previously showed snow temperate and snow height in the Appendix have been moved into the supplementary files.

– We carefully examined references to previously published work to avoid any mis- or over-interpretation and describe now more clearly how we interpret their work.

– Care was taken to clearly identify what is an assumption or hypothesis.

Additionally to these major changes, we considered all minor comments and suggestions of the Reviewer. The answers to these comments can be already found in the interactive discussion forum:

• https://www.the-cryosphere-discuss.net/tc-2019-63/#discussion

Because the manuscript was reworked substantially we think that it would not help the Reviewers to upload a track-change version. In case this is required for formal reasons or requested by the Reviewers we would definitely provide such a document.

With best regards, the authors

S. Leinss, H. Löwe et al.

[revised manuscript text omitted]

---

## Author Response (AR2)

**General comments**

Dear Reviewer #1 and Reviewer #2,

thank you very much for spending the time to carefully read the paper again. We also thank you for your constructive comments and for suggestions to further improving the paper. Please find below all remaining referee comments (RC) by Referee #2 and the corresponding author comments (AC). We thank Referee #1 already for recommendation for final publication and the Editor for excellent handling of the manuscript.

Attached to this document is the track-change version of the manuscript.

**RC1:** Specification of Amin = -0.7. This comes from extrapolation of Figure 5, yet is inconsistent with literature values of -0.3. This discrepancy is not covered in the discussion. Is this to do with the difficulty of measuring depth hoar in the field or something more fundamental in the difference between the way natural snow evolves and TGM growth in the lab?

**AC1:** The literature is not very helpful here because systematically measured values of A from laboratory experiments are simply missing, especially for time series lasting many hundreds of days. Such long time series would be required according to our model (Fig. 2a) to reach extreme anisotropy values like Amin = -0.7. As our model agrees with lab experiments lasting for about 80 days, we consider the extrapolation from Fig. 5 as a valid way to estimate $A_{\min}$ by extrapolation. Snow aged multiple hundreds of days can be found e.g. on ice sheets, however, because in nature settling occurs always combined with TGM it is very difficult to find extremely low anisotropy values. Therefore, we consider the value of Amin = -0.7 also not to be inconsistent with literature values from ice sheets, e.g. the measurements by (Fujita et al., 2014, 2016, 2009) which indicates common dielectric anisotropy of $\Delta\varepsilon = 0.05$ with extreme values up to 0.08. With the corresponding density, these measurements correspond to anisotropy values as low as A = -0.4. In addition more recent data from Greenland measured by SLF (unpublished) actually confirms Amin = -0.7 very well (see attached figure).

To consider your comment in the paper we suggest to add in Sect. 4.1 after "Therefore, we choose a pratical minimum threshold of $A_{\min} = -0.7$. "*We note that such low values have neither been observed in lab experiments nor in nature, because experiment would need to last many hundreds of days and in nature snow would either evaporate or TGM would occur combined with settling.*

**RC1.1:** What is the sensitivity of the model to Amin and Amax?

**AC1.1:** The impact is rather small and is on the same order of magnitude as the uncertainty from different ensemble runs and similar to the scatter of data points from the CT measurements. Within the range of Amin = -0.8...-0.5 and Amax = 0.2...0.4 the simulated depth-resolved, as well as depth-averaged anisotropy values change hardly more than by 0.05. Still, the dymamics of the model remain the same. Therefore we have add to Section 5.4 (model deficits) that *Varying the values of $A_{min}$ and $A_{max}$ within the estimated uncertainty range of $\pm0.1$ does not affect the general dynamics of the model. Within this range the modeled results did hardly change more than $\pm0.05$.*

**RC1.2:** If the column Amax from radar measurements is 0.2, values for individual layers could be much higher. It's not clear how Amax = 0.3 +/-0.1 was estimated. What is the impact of this estimation?

**AC1.2:** The values of the individual layers are shown in Fig. 8(b) where values of A = 0.1...0.4 have been observed for 2011-12-21 which coincides very well with the most positive column values of A =0.2 measured by radar. As the majority of CT data points in Fig. 8(b) is less than 0.3 and only a few points range to 0.4 (some even to +0.6) we think that Amax = 0.3 is a reasonable estimate. We have added to section 4.3 (validation with CT-profiles) that *The most positive anisotropy values of the CT data A = 0.3 with some scatter up to 0.4 and higher agree with our estimation of Amax = 0.3..* The impact for Amax = 0.2...0.4 is mentioned in AC1.1.

**RC2:** I'm not convinced that dividing by a density and a characteristic microstructural length scale in equation 5 on dimensional grounds is the correct thing to do. The orientational dependent part of the equation already indirectly accounts for a difference in size associated with horizontal structure vs size associated with horizontal structure so to then incorporate a size dependent term which is afterwards treated as a constant seems odd. From a physical perspective it seems intuitive that the rate of vapour-solid exchanges is dependent on the effective surface area perpendicular to the thermal gradient i.e. horizontal structures present more surface area for phase changes than vertical structures. I would then argue that you are dividing the flux by something with units of specific surface area but dependent on the anisotropy.

**AC2:** We agree that the factors introduced on dimensional grounds seem a bit odd. As we consider the density of ice $\rho_{ice}$ and the "microstructural parameter $f_\mu(\cdot)$" anyway to be constant we have now absorbed them into the prefactor $\alpha_1$ which has now the value and dimension $\alpha_1 = 1.01\,\mathrm{m^2 kg^{-1}}$. By this incorporation, all discussion about $f_\mu(\cdot)$ becomes irrelevant and has been removed. Instead we changed the first paragraph of the discussion to:

*It may surprise that we neglected any parametrization of the microstructure in the model. For example, a more sophisticated description would characterize each grain type, shape and size by its potential velocity to transform into vertical structures. The microstructure is only rudimentary considered by the factor $(A - A_{min})^2$ which causes a quadratic dependence of the change rate on the anisotropy such that*

*horizontal structures transform much faster than vertical structures.*

**RC3:** If empirical correction factor $\alpha_2$ is somehow related to the resistance of the bonded ice matrix to compression, would vertical and horizontal structures have the same value?
**AC3:** Probably not. But as we like to keep the number of free parameters as low as possible we think that using the same value of $\alpha_2$ for vertical and horizontal structures is a reasonable first guess. To mention this in the paper, we will add in section 5.4 (model deficits) *As the settling rate depends on the resistance of the bonded ice matrix to compression and as the resistance should depend on the anisotropy we think that $\alpha_2$ could also depend on the anisotropy. However, to keep the numbers of free parameters low we used constant values of $\alpha_2$.*

**RC4:** Figure 2 - it would be better to have a consistent colour scheme. At the moment red is used for (8) in (a) but (7) in (b).
**AC4:** Fine. (8) in (a) is now in dark red and the numbers correspond to the same colour.

**RC5:** Section 3.2. What frequencies / incidence angles are you using? This relates to later section and calculation of standard deviation of A from CPD in section 4.2 - how was this done? Calculated over time / frequency / incidence angle?
**AC5:** We used the full spectrum of SnowScat (9.2–17.8 GHz), split into 13 bands, each with a bandwidth of 2 GHz. As the CPD at steep (i.e small) incidence angles is rather small, we used only the incidence angle 40, 50, and 60 degree. The standard deviation was calculated over all 13 x 3 measurements with different frequencies and incidence angles. We have added this information to section 3.2.

**RC6:** Clearly a lot of work went into derivation of the meteorological data to run SNOWPACK, with associated uncertainties. Why was this approach taken rather than using the Essery https://doi.org/10.5194/gi-5-219-2016 dataset?
**AC6:** Simply because we were not aware of it because we started preprocessing the meteorological data already before 2016. The dataset contains some additional corrections which we haven't considered. We will definitely cite it (see next comment).

**RC6.1:** What could cause the incoming longwave to require a decrease (in processing in this paper) instead of providing an additional thermal contribution due to the trees?
**AC6.1:** We added at the end of section 3.3.4 (calibration and configuration of SNOWPACK) the explanation that *The reduction of short wave radiation agrees with the model by Essery et al. (2016), however, they modeled an increase of the longwave radiation by a few percent whereas we reduced it by a few percent such that SNOWPACK results agree better with snow depth and temperature measurements.*

**RC7:** P.19, 2nd paragraph - layers at the base do not always show vertical structures: at the start of the seasons there are only horizontal structures: could this period be used for calibration of Amax?
**AC7:** Yes, correct. The first snow of every season shows some settling and therefore a horizontal structures. We corrected this in the paper (always -> *mainly, except for early season snow*). For calibration of Amax from early season snow we often have the problem that the early season snow is quite wet (no radar penetration). Additionally, the depth of the first few cm of snow is pretty thin and introduces therefore a relatively high uncertainty for the anisotropy. Therefore we did not use this snow to calibrate Amax. Instead we used the radar measurements from Dec 2011 where the snow depth- and snow density related uncertainty in the anisotropy estimation is relatively low (see provided error bars / standard deviation in Fig. 6+7). See also comment AC1.2.

**RC7.1:** Similarly, could the purely vertical structure profile period in 2010-2011 be used to calibrate Amin?
**AC7.1:** We don't think so. As discussed in AC1, time series lasting many hundreds of days would be required to observes anisotropy values close to Amin. The snow in the purely vertical structure profile end of November 2010 is only about 30 days old. The profile CT-1 taken in the season 2010/2011 could provide our most negative anisotropy values, unfortunately, the strong depth hoar could not be sampled (Fig. 8a). Data from other years, e.g. CT-4 shows values of A down to -0.4 but again, the snow is only 4 months old and additional settling might have occured.

**RC8:** P.19, L9-10 (begins We think...) this sentence would benefit from some clarity
**AC8:** We rephrased this sentence to *Initially fallen crystals have an intrinsic particle orientation which is not perfectly aligned horizontally when first sticking to the surface. Upon initial metamorphism and settling, which takes some characteristic time, these crystals may further align horizontally under the influence of gravity with a measurable impact on the anisotropy.*

**RC9:** P.24, L9-10. Missing a discussion about the difference at the bottom (where observations available). The fact that the micro-CT data show vertical structures whereas the model indicates horizontal may be significant depending on application. It would be interesting to discuss possible reasons for this difference.
**AC9:** The bottom (where observations are available) would be the snow between 10 and 30 cm for which the data CT-1 indicate A = -0.1 whereas the model indicates A = 0.05. These snow layers have settled during a thin snowpack with relatively cold air temperatures (-15°C), hence large temperature gradients, where the settling rate could have been overestimated by SNOWPACK or the anisotropy evolution could have been overestimated by our model. The settling of these

[Figure]

**Figure 1.** Histogram of anisotropy measurements in Greenland (un-published data by SLF).

snow layers appears as a very fast anisotropy increase in the model results for Jan and Feb 2011 (Fig. 6d) supporting the assumption that the modeled results are too positive. Still, we think that this is a rather speculative discussion and therefore prefer to avoid it in the paper.

**RC9.1:** It is also interesting that the model agrees more with A derived from pc rather than pex for this section. What could this mean?

**AC9.1:** For CT-1 (Fig. 8a), the model seems to agree better with A from $p_c$ but we think that this is pure coincidence because the model estimates a too positive anisotropy which is closer to A from $p_c$ which is generally smaller than A from $p_{ex}$. The difference between A from $p_c$ and $p_{ex}$ is discussed in section 4.3.1. (anisotropy determined from $p_c$).

**Typos:** all listed typos have been corrected.

[revised manuscript text omitted]

---

## Author Response (AR3)

Dear Editor,

We are pleased to hear that the paper is accepted for publication. Please find attached all required files.

We did some minor editing of the manuscript to correct for typos, to improve the readability and formulated some sentences more concise. We did not change any meaning of the sentences nor any relevant information. Attached to this document is the track-change version of the manuscript.

Please note that we used the LateX package "xr" to cross-reference to the figures in the supplementary material. Therefore, the main document requires the *.aux file of the supplementary LateX document. We uploaded this file together with the figures in the zip file.

With best regards,

Silvan Leinss

[revised manuscript text omitted]